# Neuroscientific Insights into the Built Environment: A Systematic Review of Empirical Research on Indoor Environmental Quality, Physiological Dynamics, and Psychological Well-Being in Real-Life Contexts

**DOI:** 10.3390/ijerph22060824

**Published:** 2025-05-23

**Authors:** Aitana Grasso-Cladera, Maritza Arenas-Perez, Paulina Wegertseder-Martinez, Erich Vilina, Josefina Mattoli-Sanchez, Francisco J. Parada

**Affiliations:** 1Institute of Cognitive Science, Osnabrück University, 49074 Osnabrück, Germany; aitana.grasso.cladera@uni-osnabrueck.de; 2Programa de Doctorado en Psicología, Facultad de Psicología, Universidad Diego Portales, Santiago 8370109, Chile; maritza.arenas@mail.udp.cl; 3Centro de Estudios en Neurociencia Humana y Neuropsicología, Facultad de Psicología, Universidad Diego Portales, Santiago 8370109, Chile; josefina.mattoli_s@mail.udp.cl; 4Departamento de Diseño y Teoría de la Arquitectura, Facultad de Arquitectura, Construcción y Diseño, Universidad del Bío-Bío, Concepción 4051381, Chile; 5Programa de Doctorado en Neurociencias, Escuela de Medicina, Pontificia Universidad Católica de Chile, Santiago 8331150, Chile; 6Programa de Pregrado en Psicología, Facultad de Psicología, Universidad Diego Portales, Santiago 8370109, Chile

**Keywords:** indoor built environment, physiology, neuroscience, indoor environmental quality, well-being

## Abstract

The research aims to systematize the current scientific evidence on methodologies used to investigate the impact of the indoor built environment on well-being, focusing on indoor environmental quality (IEQ) variables such as thermal comfort, air quality, noise, and lighting. This systematic review adheres to the Joanna Briggs Institute framework and PRISMA guidelines to assess empirical studies that incorporate physiological measurements like heart rate, skin temperature, and brain activity, which are captured through various techniques in real-life contexts. The principal results reveal a significant interest in the relationship between the built environment and physiological as well as psychological states. For instance, thermal comfort was found to be the most commonly studied IEQ variable, affecting heart activity and skin temperature. The research also identifies the need for a shift towards using advanced technologies like Mobile Brain/Body Imaging (MoBI) for capturing real-time physiological data in natural settings. Major conclusions include the need for a multi-level, evidence-based approach that considers the dynamic interaction between the brain, body, and environment. This study advocates for the incorporation of multiple physiological signals to gain a comprehensive understanding of well-being in relation to the built environment. It also highlights gaps in current research, such as the absence of noise as a studied variable of IEQ and the need for standardized well-being assessment tools. By synthesizing these insights, the research aims to pave the way for future studies that can inform better design and policy decisions for indoor environments.

## 1. Introduction

The built environment can be defined as the natural environment modified by human conceptualizations and actions [1,2]. Considering the peculiarities of modern life, people spend approximately 90% of their time indoors [3,4,5], which makes understanding the relationship between indoor environmental characteristics and human functioning especially relevant. Researchers have explored several components of the built environment (e.g., light exposure, acoustic conditions, temperature, and air quality) to examine its relationship with physiology (e.g., cortisol levels and circadian rhythm; [6,7]), psychological states (e.g., happiness, irritability, and stress [8,9]), and cognitive processes (e.g., attention, learning, and memory; [10,11,12,13]). Findings show that some features of the environment can promote or disturb mental health and cognitive functioning [14,15,16] and can have a direct impact on neurobiology [17].

### 1.1. The Built Environment’s Impact on Health

The built indoor environment can influence mental health and well-being through both negative and positive environmental features. Several studies have shown that certain stressors commonly found in indoor settings, such as crowding, noise, and environments perceived as unsafe, may negatively affect psychological states, leading to chronic stress and symptoms associated with psychiatric conditions [18,19,20,21].

In contrast, other characteristics associated with indoor spaces or their immediate surroundings have been shown to foster psychological well-being. For example, having green views from indoors, such as visual access to trees or vegetation through windows, has been linked to increased perceived well-being and reduced mental fatigue [8,22,23]. Similarly, exposure to natural light within indoor spaces may alleviate symptoms of seasonal depression [19,24,25]. Overall, improvements in the indoor environment have been associated with reduced physiological and psychological distress in various contexts, including workplaces and healthcare settings [6,26,27,28,29,30,31].

The framework commonly used to conceptualize these effects is indoor environmental quality (IEQ), which refers to the set of physical conditions in indoor spaces that affect occupants’ health, comfort, and well-being [32,33,34]. According to systematic reviews, IEQ dimensions such as air quality, thermal comfort, acoustic conditions, and lighting are considered positive when they help reduce discomfort or promote cognitive and emotional functioning [35,36]. These factors have been studied in various contexts. For example, Turunen and colleagues [37] found associations between classroom noise and poor ventilation and symptoms like fatigue and headaches in students. Similarly, Salonen and colleagues [36] showed that adequate ventilation and favorable visual conditions, such as appropriate lighting levels, can positively affect the well-being of individuals in healthcare environments.

### 1.2. A New Perspective for Neuroscience Research of the Built Environment

An alternative paradigm to classic conceptualizations in cognitive science conceives the nature of the cognitive process as a complex phenomenon that emerges from the dynamic relationship between the brain/body system of an agent in active interaction with its environment [38,39,40,41]. Under this paradigm, cognition is understood as an embodied, environmentally scaffolded, and enactive process. This means (1) that the brain and all the agent’s biology play a major role in cognition (i.e., embodied) [41,42], (2) that cognitive processes are dependent on the environment and also have been transformed by environmental resources (i.e., environmentally scaffolded) [43,44,45], and (3) that the mind is the product of the dynamic relationship between brain/body and environment (i.e., enactive) [42,46]. Hence, the consideration of intracranial (e.g., brain activity) and extracranial dynamics (e.g., body and environment) allows for the understanding of cognitive processes as embodied in biology and scaffolded by the environment [43,45,47,48,49,50,51,52]. This paradigm to understand and study cognition has been conceptualized as the 3E cognition perspective or 3E-Cognition (The revision of the 3E-Cognition paradigm exceeds the aim of the present review we kindly direct readers to [40,41,43,44,53]). In the context of this review and under the 3E-Cognition framework, well-being is conceptualized as a dynamic, embodied state arising from the continuous interaction between an individual’s physiological, emotional, and cognitive processes and their environmental scaffolds [53]. Well-being reflects the agent’s capacity to maintain adaptive regulation of bodily states, affective experiences, and cognitive engagement within specific environments. It is not a static condition, but an enacted and environmentally supported process that promotes physical comfort, emotional resilience, cognitive clarity, and effective action in the world.

These new ideas allow for the development of new research questions and hypotheses about the relationships between the body, brain, mind, and environment [41,54]. Hence, this perspective can be considered an effort to generate a multi-level evidence-based approach for studying this dynamic system composed of the brain and the body in its interaction with the environment and its implication for cognition, psychological status, and even social interactions.

Considering the aforementioned concepts, important questions arise regarding how different components of the built environment influence physiological responses. To illustrate existing methodological efforts in this area, we highlight examples from the literature on thermal environmental changes and their associations with physiological responses, such as body temperature regulation and its relationship to brain and cardiac activity. These studies exemplify how researchers are beginning to approach IEQ assessments through an inclusive lens that integrates physiological and environmental data. For instance, the impact of the thermal characteristics of the environment on the body has been explored in previous research due to the connection between vagal and sympathetic nerve activity and its role in thermoregulation [55]. Hence, thermal comfort has been studied in relation to its impact on heart activity, showing that a higher ratio of low frequency/high frequency relates to unpleasant thermal sensation and discomfort [56,57,58]. Similarly, Mulders and colleagues [59] assessed brain activity concerning thermal stimuli, showing differences in terms of time–frequency analyses. Furthermore, alpha activity during exposure to natural settings presented similar characteristics to alpha activity during relaxation and restoration states regarding alpha–theta oscillations and synchronization [60,61].

### 1.3. Methodological Implications for Neuroscience Research

Research from the neuroscience field -broadly understood as the investigation of brain and body dynamics that directly or indirectly reflect cognitive, affective, and physiological processes- has incipiently addressed the relationship between environmental variables, cognitive processes, and physiological states using different methods. For instance, some studies have utilized real-time biometric data (e.g., electrodermal activity, EDA, and heart rate, HR) as a measure of physiological reactions to the built environment [56,62], electric brain activity (e.g., via electroencephalogram, EEG) to assess the impact of the built environment’s attributes on brain activity [63,64,65,66], and the implementation of self-report measures as the most common technique to collect information from participants [3,63,67]. Table 1 provides a map on different physiological measures, its relevance, and the degree of directness to cognitive processes.

As shown by Azzazy and colleagues [3], the study of brain activity has remained to classical paradigms (e.g., picture presentations [63]) and non-mobile techniques for data acquisition (e.g., functional magnetic resonance imaging, fMRI, and computerized tomography, CT). Only during the past decade have researchers changed paradigms to study brain dynamics in the built environment using virtual reality [64,65,68] or in real-world settings [3,66,69,70,71,72].

These findings are consistent with new technological advancements of the last 15 years related to the Mobile Brain/Body Imaging (MoBI) framework, which allows for the measuring of different body signals during natural movement [73,74,75,76]. This technical–methodological approach has increased the possibility of studying physiological variables in ecologically valid paradigms and real-world situations as they naturally unfold [54,75,77,78,79,80]. By doing so, it is possible to embrace the complexity of the relationship between the body, mind, and environment and address a broader understanding of their relationship and physiological dynamics as they naturally unfold in their interaction with the built environment.

### 1.4. The Present Review

This review will focus on the IEQ variables since they have been defined as a proxy for environmental quality and comfort, which are closely related to well-being. Previous reviews have already addressed the relationship between the built environment and well-being from a neuroscientific perspective [3,54,81]. However, the present review is situated from a 3E perspective, so it aims to incorporate not only brain activity but also multiple physiological signals in the study of the impact of the indoor built environment on psychological well-being, as well as a technical–methodological perspective for analyzing the existent literature.

In this sense, the present review aims to systematize the current scientific evidence on methodologies used to investigate the impact of the indoor built environment on well-being, focusing on assessing physiological variables. By doing so, we intend to (1) categorize the main IEQ variables studied in research investigating the impact of the built indoor environment on well-being in real-life settings, (2) identify and summarize methodological aspects (e.g., data collection techniques, study settings, and biomarkers) used to explore the relationship between the indoor built environment and well-being, and (3) review the self-report instruments employed to capture subjective experiences related to indoor environmental quality and well-being.

## 2. Methodology

### 2.1. Protocol and Registration

This systematic review was conducted following the Joanna Briggs Institute (JBI) guidelines for systematic reviews [82,83] and following the Preferred Reporting Items for Systematic Reviews and Meta-Analyses (PRISMA) guidelines [84]. The protocol can be found at the Open Science Framework (OSF; https://osf.io/qc5tx/ (accessed on 19 May 2025)).

### 2.2. Eligibility Criteria

Since the present review aims to systematize the current scientific evidence on methodologies used to investigate the impact of the indoor built environment on well-being, with an emphasis on the assessment of physiological variables, only empirical studies on indoor built environment quality, incorporating physiological and well-being self-report measurements, were assessed for eligibility.

To be included in the present review, studies should (1) have neurotypical human participants; (2) study one or more of the four variables related to indoor environmental quality (i.e., air quality, thermal comfort, noise, and lighting); (3) be an empirical article that addresses the relation with, at least, one physiological signal; studies only incorporating behavioral measures (e.g., only implementing eye tracking or motion energy analysis) were not considered for eligibility since the main purpose of this review is to assess physiological variables; and (4) incorporate at least one self-reported measure of well-being. Furthermore, studies will only be eligible if they are conducted in real-world scenarios (e.g., real office or residential environments or a space accommodated to simulate the characteristics of those scenarios) instead of classical laboratory setups (i.e., participants performing a task with lower degrees of ecological validity), which is understood as the limited representational fidelity of the stimuli and an insufficient alignment of tasks with functional, goal-directed behavior.

All experimental contexts were included in this review. All available publications in English or Spanish were eligible for inclusion, and no temporal limits (i.e., publication year) were applied at any stage of the search or selection process. All observational and experimental designs were considered, and no gray literature (i.e., reports that have not been included in a peer review process) was included. Table 2 provides an overview of the inclusion and exclusion criteria applied to the articles.

### 2.3. Information Sources

For this review, we conducted a comprehensive literature search on electronic databases in Web of Science (WOS), PUBMED, and SCOPUS. The reference list of key articles and reviews was screened for additional studies [3,54,85,86]. All databases were consulted in October 2024.

### 2.4. Search

A search strategy was developed following the Peer Review of Electronic Search Strategies (PRESS) checklist [87]. The search strategy was adapted to each database. The search process was conducted by two members of the research team (AGC and MAP) after a preliminary, non-systematic review of keywords related to indoor environmental quality. This step was conducted without the collaboration of a librarian due to institutional limitations. The search strategy aimed to identify studies exploring the relationship between indoor environmental quality (IEQ) and physiological or neurophysiological responses. To that end, each IEQ-related term—“indoor thermal comfort”, “indoor air quality”, “indoor noise”, “indoor light”, and “indoor environment”—was combined individually with each physiological keyword: “physiolog*”, “heart*”, “brain”, “respiration”, and “skin temperature”. For example, one of the search strings was “indoor air quality” AND “physiolog*”. A full list of search strings is available in the study protocol. Keywords were searched in the articles’ abstracts. Systematics reviews and conference abstracts were excluded from the search.

### 2.5. Selection of Sources of Evidence

The search process results were exported into Microsoft Excel [88], and all duplicated articles were removed. Four independent reviewers (AGC, MAP, EV, and JMS) performed the screening process for article inclusion. First, the reviewers selected articles based on the articles’ titles and then based on abstract information. For each stage, a consistency analysis was conducted to determine the level of agreement regarding eligibility criteria. After achieving ~90% agreement, each reviewer selected articles independently. Disagreements between reviewers were addressed through multiple rounds of discussions.

### 2.6. Data Charting Process

The reviewer team developed a Google Forms questionnaire for the data charting process. The questionnaire asked questions about article descriptions and items inspired by the present review’s goals. To ensure internal consistency, four authors (AGC, MAP, EV, and JMS) codified the first 10 articles, and the rest were divided equally and were reviewed independently by the same researchers.

### 2.7. Data Items

The questionnaire’s items referred to (1) the article’s characterization (e.g., year of publication and country) and (2) the methodological considerations (e.g., aim, environmental variable, data collection technique, setting, and task, among others).

### 2.8. Risk of Bias Estimation

The potential risk of bias in the studies was assessed using the JBI Critical Appraisal Tools [89,90], with the specific tool extension applied based on the study design of each included article. Two authors conducted this process independently (MAP and EV).

### 2.9. Strategy for Synthesis of Results

Following the narrative methodology proposed by Arksey and O’Malley [91], data were summarized following the data extraction categories aimed at answering this review’s objectives and questions. Data are presented graphically (graphs or diagrams) and in tables. Due to the high heterogeneity of the included articles, mainly in terms of the main goal, experimental design, and measured outcomes, a meta-analysis cannot be performed [92,93].

## 3. Results

### 3.1. Study Selection

After conducting the search process in all three databases, 2701 articles were found. After removing duplicated articles, 1562 articles were left. To select the articles, we conducted a two-stage screening process. Based on the information presented in the title and abstract, 1189 articles were excluded. After the full-text review, a total of 15 articles were included in the present review.

Among the principal reasons for exclusion are (1) only the use of self-report measures to study an IEQ variable (*n* = 117), (2) studying a non-target characteristic of indoor environmental quality (*n* = 79), (3) only the use of behavioral measures (*n* = 60), and (4) studies conducted in traditional laboratory settings (*n* = 49).

A total of 12 other articles were found manually using existing reviews’ citations. These reports excluded the following: eight articles due to being duplicates and four articles which focused on non-target characteristics of indoor environmental quality. Figure 1 shows the flow of articles across the different stages.

### 3.2. Study Characteristics

Although no time restriction was established a priori, all included articles were published between 2018 and 2024. The results show that 11 out of 15 articles (73%) were conducted in the last four years (Figure 2). Regarding geographical location, there is a predominant presence of studies conducted in the Asian continent, with eight of them performed in China [57,94,95,96,97,98,99,100]. Switzerland follows with two articles [101,102], while the remaining articles were published in various other countries, including Italy [103], the USA [104], Sweden [105], India [106], and England [107]. Interestingly, all included studies were conducted in the Northern Hemisphere, showing a lack of research from southern parts of the world (Figure 3).

### 3.3. Risk of Bias

The results of the risk-of-bias assessment are summarized in Table 3. Each study included in this review was evaluated using the design-specific JBI Critical Appraisal Tool. The 13 case series studies met most of the criteria outlined by the tool; however, in all of them, it remained unclear whether participants were consecutively included.

Regarding the cross-sectional studies, both met six out of the eight criteria analyzed. However, the articles lacked sufficient information to clearly address items related to identifying and managing potential confounders. 

### 3.4. Synthesis of Results

#### 3.4.1. Aim

Regarding the main aim of the articles included in this review, it is possible to notice a diversity of objectives pursued. This mainly refers to the development of different hypotheses regarding the impact of the indoor built environment on physiology and well-being/cognition. Despite this diversity, the main objective is to study the effects of aspects of the indoor built environment on physiology and cognition. It is important to notice that the majority of studies (12 out of 15) manipulated at least one IEQ parameter, most commonly temperature (*n* = 8), lighting (*n* = 5), or air quality (*n* = 5). Table 4 displays the main objective of each included article in this review.

#### 3.4.2. Indoor Environmental Quality Variable

The IEQ construct involves four main dimensions: thermal comfort, air quality, noise, and lighting. As shown in Table 4, this review identified thermal comfort as the most common dimension studied (*n* = 11) [57,95,96,98,99,100,101,102,103,104], followed by lighting (*n* = 5) [94,98,102,105,106] and air quality (*n* = 3) [98,103,107]. Noise was not explicitly addressed in the studies reviewed, highlighting a gap in research on this dimension.

Only three studies (20%) considered more than one variable of the IEQ constructs in their research questions [98,102,103], emphasizing a tendency to focus on isolated aspects of the indoor environment rather than adopting a holistic perspective. Notably, these three studies consider thermal comfort along with another IEQ factor.

Finally, the included studies exhibited substantial variability in sample sizes, ranging from 10 to 2110 participants. Most experimental studies involved small to moderate samples (*n* = 10–84), while only two field studies [96,99] incorporated larger samples exceeding 200 participants. This distribution reflects the predominance of controlled laboratory designs in the current research landscape, alongside a smaller but important contribution from large-scale observational investigations.

#### 3.4.3. Task

The reviewed articles described a diversity of tasks during environmental manipulation and physiological recording. The most common types of tasks consisted of office-related tasks or similar activities, such as reading or drawing (*n* = 8) [94,100,101,102,104,105,106]. Neuropsychological assessments, like the Stroop test, were also frequently used (*n* = 4) [98,103,107,108]. Less common tasks included physical activity (*n* = 1) [95] and sleeping (*n* = 1) [97]. Additionally, one study did not specify the nature of the task performed by participants [99]. Table 5 summarizes the tasks performed during measurements in the included articles.

#### 3.4.4. Setting/Context

This review focused exclusively on studies conducted in real-life contexts (Table 5). The most common settings were educational or university environments (*n* = 7) [94,96,99,103,105,106,108], followed by professional or workplace settings, specifically offices and office-like environments (*n* = 6) [98,100,101,102,104,107]. Additionally, residential contexts (*n* = 1) [97] and a physical training center (*n* = 1) [95] were included.

#### 3.4.5. Data Collection Technique

The reviewed articles employed various data collection techniques, depending on the physiological variables of interest. For cardiac activity, the most common methods included heart rate monitors (*n* = 6) [94,95,98,101,103,105] and the use of sphygmomanometers (*n* = 3) [98,106,108], pulse oximeters (*n* = 3) [96,106,107], and blood pressure monitors (*n* = 2) [95,99]. Regarding skin temperature, surface thermometers were the most frequently used devices (*n* = 10) [95,96,97,99,100,101,102,104,107,108]. An infrared imager was employed in one study [95]. Only one article collected information on electrical brain activity using electroencephalography (EEG) [107]. Table 6 summarizes the instruments used and measurements conducted in the included articles.

#### 3.4.6. Biomarker

In the reviewed articles, the analyzed biomarker was restricted to the type of physiological signal collected (Table 6). The most common units of analysis for cardiac activity were heart rate (*n* = 10) [95,96,98,99,101,103,105,106,107,108] and blood pressure (*n* = 5) [95,98,99,106,108]. For temperature-related measurements, the main biomarker was skin temperature (*n* = 9) [64,95,97,99,100,101,102,104,107]. The most frequently analyzed parameter regarding respiratory activity was respiratory rate (*n* = 2) [103,107]. Finally, the study that implemented EEG as a data collection technique performed frequency band analysis [107].

From the total of included articles, five studies analyzed only one biomarker [94,97,100,102,104], while ten studies incorporated two or more biomarkers [64,95,98,99,101,103,105,106,107,108].

#### 3.4.7. Well-Being Self-Report Measure

The reviewed studies employed a wide range of self-report methods, reflecting the diversity in their objectives and approaches. As shown in Table 6, the most commonly used tools included the thermal comfort vote (TCV), thermal sensation vote (TSV), and other thermal perceptions. The regulations indicated by the American Society of Heating, Refrigerating, and Air-Conditioning Engineers (ASHRAE) were used as guidelines for a number of articles in this review. In addition to thermal-related measures, some studies incorporated tools to evaluate emotional states, such as the Positive and Negative Affect Schedule (PANAS), or subjective responses like the Stanford Sleepiness Scale and the Self-Assessment Manikin (SAM) for emotional reactions.

A closer examination of the self-report instruments used across the reviewed studies reveals both convergences and divergences in methodological choices. A substantial proportion of the articles relied on scales derived from ASHRAE guidelines, reflecting a shared emphasis on thermal-related perceptions. This convergence suggests a partial standardization within the field, at least regarding thermal comfort assessments. However, significant variability was also observed. Some studies broadened their focus to include emotional states [94,107] or more comprehensive measures of subjective experience, such as the profile of mood states (POMS) or the Self-Assessment Manikin (SAM). Moreover, while several articles limited their evaluations to thermal parameters, others incorporated perceptions of lighting [106], air quality [98], and overall visual comfort [105]. These differences highlight a lack of unified criteria for self-report measurement selection, reflecting both the multidisciplinary nature of the field and the ongoing challenges in operationalizing the construct of well-being in relation to the built environment.

Also, it is important to notice that the scales/questionnaires included were not exclusively based on well-being (e.g., psychological well-being). Most of these instruments were primarily focused on assessing participants’ comfort and subjective experience regarding environmental conditions. Furthermore, only six studies explicitly conducted a cross-test between physiological responses and self-report measurements. This observation highlights a frequent methodological gap: while many studies simultaneously collected physiological and subjective data, fewer statistically analyzed their relationship.

## 4. Discussion

The present literature review systematized the current scientific evidence on methodologies used to investigate the impact of the indoor built environment on well-being, focusing on assessing physiological variables (e.g., cerebral and cardiac activity and skin conductance). After completing the systematic searches, 15 articles matched our criteria for inclusion. In the following sections, we will discuss what we consider to be some of the most relevant aspects for advancing research in this field.

### 4.1. Research Field Characterization

From the general analysis of the research characteristics, it is possible to notice a significant increase in studies conducted in the field over the past four years (*n* = 11), accounting for more than 70% of the included studies. Several factors may have contributed to this trend, including the general rise in scientific publications across disciplines, technological advancements that have facilitated real-world physiological measurements, and growing attention to the role of indoor environments in health. Additionally, it is possible that the COVID-19 pandemic further increased the amount of time people spent at home, intensifying exposure to residential indoor environments compared to other indoor settings such as workplaces, schools, or public buildings [109,110,111] and contributing to this renewed focus. While this interpretation remains tentative, it highlights how scientific inquiry is deeply influenced by socio-cultural, political, and economic realities [112,113]. In this sense, although the aspiration for scientific knowledge is generalizability, it remains crucial to acknowledge the specificity of contexts and the need for science to remain sensitive to diversity.

### 4.2. Indoor Environmental Quality Variables

From the results presented, it can be understood that the current models that govern building designs are based on the simplistic assumption that human beings react in a disjointed and monotonous way to the stimuli they are exposed to. To evaluate the comfort and well-being of people in indoor spaces, the research has addressed the assessment of IEQ factors. This has mainly been performed separately. Only 3 out of the 15 studies link more than one factor [98,102,103], while only 1 study considers three factors [98].

None of the eligible articles in this systematic review considered noise as an IEQ variable to be studied, a finding that stands out given the established role of noise (or unwanted sound) as a risk factor capable of directly or indirectly activating stress pathways, potentially affecting health and well-being [114]. This omission is notable, as research demonstrates noise exposure’s adverse physiological and psychological effects [115,116,117]. A possible explanation for this gap may lie in the predominant focus on thermal comfort and energy efficiency observed in the articles included in this review, suggesting a prioritization of these aspects over other critical environmental factors such as acoustic quality. This underscores the need for future studies to adopt a more integrated approach to indoor environmental quality by examining noise alongside thermal comfort, lighting, and air quality. A holistic understanding of well-being requires considering how these factors interact, both within individuals and across different occupants, rather than assessing them in isolation.

The reviewed articles lack a holistic vision of IEQ criteria, focusing mainly on thermal comfort, which appears to be the most influential factor in user performance and behavior [118]. However, this focus often overlooks the integration of other relevant environmental stimuli. This integration between the senses and their information content is associated with multi-modal phenomena and cross-modal stimuli interactions. The former comprise a combination or union of multiple unisensory and independent inputs, while the term cross-modal refers to situations where a stimulus of one sensory modality is shown to exert an influence on our perception or responsiveness to stimuli presented in another sensory modality [119].

The built environment is a complex system characterized by feedback, agent interrelation, and non-linear, discontinuous relationships. However, the empirical results that support its development often focus on objective associations between stimuli and responses without appreciating the complexity of built environments. These contexts involve combinations of continuous and transient exposures and produce multi-layered psychophysiological effects that drive the occupant’s perception and behavior [120,121]. Determining the cause-and-effect relationships between well-being, comfort, and environmental parameters is complex because the combinations, apart from being multiple, can interact synergistically or cancel each other out antagonistically, influencing the occupants’ physical, physiological, and psychological responses [122].

Nevertheless, human responses to indoor environmental quality (IEQ) are inherently multisensory and shaped by the dynamic integration of multiple environmental cues. Rather than being driven by isolated parameters, psychological and physiological responses often emerge from the interplay between sensory modalities such as thermal, visual, olfactory, and auditory stimuli. For instance, studies have shown that the perception of thermal comfort can be modulated by lighting conditions, with natural light enhancing tolerance to warmer temperatures [68,123]. Similarly, air quality may interact with thermal environments to influence cognitive performance and perceived well-being [124,125]. This highlights the need for future research to move beyond unidimensional models and develop composite IEQ frameworks that account for interaction effects, e.g., thermal × natural light or thermal comfort × air quality, to better capture the complexity of real-life environmental experiences. Such integrative approaches are particularly crucial in neuroscientific investigations where multisensory convergence plays a key role in shaping emotional, cognitive, and behavioral outcomes [126]. Embracing this complexity will enhance built environment research’s ecological validity and translational value.

Further research that considers the simultaneous interrelation and integration between environmental factors and their effects on people from the technical, social, mental, and physiological construct is needed [127,128]. Neither the perception of comfort nor that of well-being in the built environment can be extended linearly from one physical domain to another, and although the evaluation becomes more complex, it must be taken into account that stimuli continuously and interrelatedly influence personal evaluations.

Moreover, it is worth noting that several global healthy building frameworks, such as the WELL Building Standard [129], have explicitly aligned indoor environmental factors with human health, comfort, and well-being outcomes. For example, lighting conditions are evaluated using metrics like Equivalent Melanopic Lux (EML) to account for circadian and physiological effects. Recent studies have shown that WELL-certified and LEED-certified buildings tend to exhibit improved indoor environmental quality parameters compared to non-certified buildings, highlighting the tangible link between building standards and occupant well-being [130]. Additionally, certifications such as WELL incorporate multi-criteria evaluation methods that reflect a holistic view of human well-being in relation to the built environment [131]. The growing alignment between healthy building certifications and scientific research on IEQ underscores the importance of adopting interdisciplinary and ecologically valid approaches when assessing the impacts of the built environment on occupants.

### 4.3. What Does Well-Being Mean

The methods used in the reviewed articles to assess self-reported well-being are highly diverse. Most commonly, they involve measures of satisfaction and/or comfort derived from ASHRAE standards, such as the thermal sensation vote (TSV) and thermal comfort vote (TCV). This variability may reflect the lack of consensus on conceptualizations of well-being.

Well-being is widely recognized as a complex and multidimensional construct that integrates psychological, social, and physical dimensions and is influenced by the environment in which individuals find themselves [115,132,133,134]. It can be understood as a dynamic state of equilibrium that may be affected by life events and challenges, encompassing dimensions such as emotional fulfillment, psychological health, and social connections [135]. However, there is no international consensus on its definition, and different disciplines conceptualize well-being in varied ways. Often, the term is used simplistically as synonymous with *wellness*, *happiness*, and *quality of life* or is associated with *comfort* and *health* [136,137,138,139].

Similarly, there is no clear consensus on what constitutes well-being in relation to the built environment. Various definitions and conceptual frameworks have been proposed, often influenced by disciplines such as architecture, environmental psychology, and public health [115]. The complexity arises because well-being in built spaces can be affected by a broad range of factors, including environmental stimuli, individual preferences, and broader societal norms [18,132]. Moreover, tensions may emerge between architectural goals, such as energy efficiency and the well-being of occupants, indicating that achieving a balance is challenging and highly context-dependent [35]. Due to the lack of consensus on the definition of well-being in the built environment, there is no straightforward approach to measuring well-being in existing or new buildings. Some studies propose that well-being should be assessed through both subjective and objective measures, acknowledging its holistic nature while differentiating it from related concepts such as health and quality of life [132,140]. Current methods typically involve medical examinations, extensive self-report questionnaires (not exclusively focused on well-being), and diverse observational and monitoring techniques [136], though these approaches rarely incorporate comparative analysis with neurological variables.

On the other hand, and as mentioned, it is extremely complex to broadly measure well-being due to the complexity of its definition that addresses states related to health, other perceptions according to immediate requirements (i.e., comfort), and other subjective and emotional terms such as love or happiness [139]. To respond to the needs and requirements of building users for both temporary comfort and sustained well-being, it is important to advance the definition of parameters and their interactions, including potential compensations between them. This progress will enable us to characterize people’s responses to this multi-modal and cross-modal environmental stimulation and evaluate how these responses and the corresponding behaviors change over time, depending on the context and personal characteristics of the individual. Achieving this, however, requires establishing a clear and consensual framework for what should be tracked and measured. Progress must be made towards a paradigm shift from current methods and metrics that are typically and repeatedly used to evaluate the qualities and performances of a built environment for (or towards) understanding intermodal stimulation and the synergistic integration between different indoor environmental quality parameters from an interdisciplinary perspective.

### 4.4. Physiology: What Should We Be Looking for

Considering the diversity of aims, experimental designs, and technical–methodological aspects of the reviewed studies, it is difficult to determine one exclusive biomarker to address the study of the impact of the indoor built environment on well-being. Our results indicate that the type of physiological analysis employed in each study is largely dictated by the chosen data collection technique and the specific research question under investigation. For instance, *iButton* devices (Analog Devices, Inc. (ADI), Wilmington, MA, USA) are commonly used for skin temperature measurements, ECG for heart rate and heart rate variability, and EEG for frequency-based analyses of brain activity. However, these physiological signals have largely been treated as isolated measures, with little consideration of their simultaneous interactions within a broader embodied system.

To critically advance the field, we must move beyond fragmented physiological analyses and toward a more integrated understanding of bodily dynamics. Indoor built environment research could benefit from studying physiological activity in light of brain/body physiological dynamics and interoceptive processes (e.g., using heartbeat-evoked potentials (HEPs) [141], movement-related potentials (MRPs) [142], blink-related potentials (BRPs) [143], and fixation-related potentials (FRPs) [144]). A clear example is the case of HEP, which has been widely implemented for the study of interoception [145], a process that has been used for assessing well-being [146,147,148]. Hence, implementing these types of integrative analyses might be useful in understanding bodily dynamics modulated by specific aspects of the indoor built environment.

However, the inclusion of these complex analyses is not exempt from challenges. To be properly included, research design and analysis changes might be necessary. In this sense, clear hypothesis-driven analytic strategies based on appropriate theoretical frameworks will be required to reduce the dimensionality of potentially unlimited variables [80,149]. An example of how researchers can overcome this challenge is by implementing the *scalable experimental design* (SED) heuristic [78,150,151], which will allow for the design and development of classically structured experiments that later can be expanded towards real/virtual environments. One of the main premises of SED is that this type of experimental design will allow for the parametric testing of reliable neurobehavioral markers (e.g., HEP) in different settings, from laboratory to everyday situations. In short, SED posits the opportunity to study the evolutionary-given sensorimotor possibilities of the encounter between the agent and the environment. This might be relevant for the research field of indoor built environments since it will allow for the development of specific research questions regarding the impact of indoor variables, as they naturally occur in the real world in cognitive (e.g., attention), psychological (e.g., well-being), and even interactional (e.g., environment as a facilitator for social interaction) processes that can be carefully and systematically tested in research. Nevertheless, technical–methodological development is needed to move forward in the research field. Even when (neuro) physiological markers or dynamics might be identified and systematically studied, there is still a need for working within a coherent and unified theoretical framework to generate connections between empirical data and the existing knowledge and theories [152].

Another important issue regarding physiological analysis is the necessity of interdisciplinary research teams or collaborations. The importance of having experts from different research fields who can contribute to the analyses and interpretation of the collected data, especially due to the lack of standardized biomarkers for the study field, is increasing. Specifically, including neuroscientists and cognitive scientists will allow us to deal with large and complex databases and move forward in identifying key biomarkers or physiological components to parametrically and systematically incorporate and explore them from classical laboratory settings to the real world.

### 4.5. From Plan Drawing to the Real World

Research on the indoor built environment seeks to understand how environmental variables shape diverse aspects of human life: daily activities (e.g., work, study, and rest), leisure, and health, among others. Understanding these variables can help us incorporate them into architectural design. In this sense, we identify at least two key elements to consider before conducting research.

First, there is a need to implement research paradigms that allow for the study of everyday dynamics [153]. Designing and incorporating experimental tasks resembling real-world tasks and demands are crucial. From the articles included in this review, the realization of neuropsychological assessments in contexts with different indoor environmental variables being manipulated (e.g., luminosity and air ventilation) was the most common experimental tasks. Even when these are specific and structured tests designed for studying cognitive functioning, it is possible to test how cognitive functioning might be affected by environmental variables since they resemble cognitive load in, for example, everyday work or study-related contexts.

On the other hand, the incorporation of naturalistic research paradigms that emulate real-world activities and neurobehavioral processes with higher degrees of ecological validity (i.e., less restrictive settings than the classical neurocognitive laboratory) can contribute to the understanding of the impact of the environment on cognitive and physiological processes as they naturally take place in the world. Here, the SED approach described before offers the possibility to first develop well-controlled laboratory studies to test experimental factors despite their diminished degree of ecological validity [150]. Then, it will provide the chance to conduct real-world experiments with less experimental control but with higher ecological validity regarding action and behavior [150,152]. As elegantly outlined by Gramann and collaborators [152,154], the real world turns *into the* laboratory. Regarding our results, some of the included articles implemented naturalistic types of tasks (e.g., book reading and office tasks), which show the intention and motivation for generating insights regarding real-world contexts.

Considering the interest in conducting research with valid and meaningful questions to contribute to the design of the indoor built environment, the development of new technologies for research could not be conducted at a better time. Almost two decades ago, the Mobile Brain/Body Imaging (MoBI) technical–methodological framework offers the possibility to move the neuroscientific research field from the laboratory to the real world, thanks to the portability of devices and the progress made in terms of managing large and complex data analyses [73,155,156,157]. MoBI’s portability and advances in real-time data acquisition and processing enable researchers to track cognitive, physiological, and behavioral responses as individuals engage with their surroundings naturally. Hence, by achieving greater theoretical–methodological coherence, the field can move toward integrative, multi-modal research approaches that capture the dynamic, embodied, and situated nature of cognition. Ultimately, this will generate deeper insights into the ways built environments shape cognitive processes, psychological well-being, and health, paving the way for truly evidence-based architectural and environmental design [153].

MoBI represents a methodological advancement uniquely suited to addressing key gaps identified across the reviewed literature [158]. Specifically, MoBI enables the simultaneous recording of neural, behavioral, and bodily dynamics in real-world or ecologically valid environments. This approach directly aligns with the embodied, environmentally scaffolded, and enacted principles of the 3E-Cognition framework that grounds this review. While traditional laboratory-based neuroscience often isolates cognitive processes from natural contexts, MoBI captures the brain–body–environment system as it dynamically unfolds [157]. The advantages of MoBI include the ability to study active cognition and behavior outside of constrained laboratory settings [150], providing critical insights into brain–body coupling mechanisms during naturalistic interactions [159,160,161]. Moreover, it allows researchers to explore how environmental features shape neurobehavioral dynamics, a dimension that static or purely lab-based methods cannot adequately capture.

Nonetheless, it is important to acknowledge MoBI’s current limitations, including the potential presence of motion artifacts [162], increased data complexity, and the technological challenges associated with portable neuroimaging. These issues, however, are being progressively addressed through advances in signal processing, hardware design, and hybrid methodological frameworks.

Thus, our recommendation for the increased use of MoBI stems not from personal preference but from the necessity to implement more dynamic, context-sensitive methodologies that are theoretically and empirically suited to studying cognition as it naturally occurs in lived environments.

## 5. Limitations

This systematic review presents some limitations that must be considered when interpreting the results. Importantly, these limitations arise from the current developmental stage of the field and the inherent characteristics of the included studies, rather than the methodological approach of the review itself. While our review followed established systematic guidelines and aimed for methodological transparency, we acknowledge that the specificity of our search strategy may have unintentionally limited the breadth of the included studies. In particular, the requirement to combine the term “indoor” with each indoor environmental quality (IEQ) domain (e.g., “indoor noise” and “indoor lighting”) may have excluded relevant studies that examine environmental conditions within buildings but do not explicitly use the term “indoor”. Furthermore, the inclusion criteria applied inconsistently across domains; for example, “thermal comfort” was used as a compound term, while other domains like “air quality”, “noise”, and “light” were included in simpler forms. Additionally, we did not use wildcards or lexical variants (e.g., “light*” to capture “light” and “lighting” or “sound” and “acoustic” as alternatives to “noise”), which may have further reduced the number of retrieved articles. As a result, the relatively low number of included studies in certain domains may not accurately reflect the overall research activity in the field, but rather a consequence of our constrained lexical strategy. We see this as a critical limitation of the current review and encourage future systematic reviews in this domain to employ broader, more inclusive term sets and consider piloting alternative search strategies to improve coverage and sensitivity.

Furthermore, no eligible articles in the reviewed literature included noise as an indoor environmental quality (IEQ) variable, despite its recognized importance in influencing well-being and comfort in indoor environments. This gap highlights the need for more comprehensive research incorporating noise alongside other IEQ variables to better understand its impact on physiological and psychological states. However, it is important to consider that the conceptualization of noise for the searches might have been a limitation on the number and type of articles.

In addition, while our focus on ecological validity aimed to capture studies conducted in real-world contexts, we recognize that the broader concept of external validity, particularly the generalizability of findings across populations, settings, and cultures, may be limited. The heterogeneity of study designs, participant characteristics, and contextual variables in the included studies poses challenges for drawing generalized conclusions. This limitation further supports the need for standardized methodologies and cross-context replications in future research.

Also, the studies included in this review were predominantly conducted in the Northern Hemisphere, with no representation from the Southern Hemisphere. This geographical limitation restricts the applicability of findings to regions with different cultural, environmental, and climatic conditions. Third, most studies focused on isolated IEQ factors, such as thermal comfort or lighting, rather than addressing the dynamic interplay between multiple environmental stimuli. This fragmented approach limits the understanding of multi-modal and cross-modal effects that are critical for capturing the complexity of human–environment interactions. Lastly, the wide range of self-report methods reported introduces variability in definitions and measurements, complicating the comparability of findings.

## 6. Conclusions

The present systematic review highlights methodological aspects of the indoor built environment research field in combination with neuroscience research. The theoretical-methodological development in neuroscience research (i.e., 3E Cognition/Mobile Brain/Body Imaging framework) can provide insights to generate new research questions, as well as collect, analyze, and interpret larger datasets to comprehend the impact of the indoor build environment on cognition, psychological well-being, and health.

Notably, across the reviewed literature, few studies employed explicit hypothesis-driven experimental designs, with most adopting exploratory or descriptive frameworks. This highlights a methodological gap that future research should address. We recommend that subsequent studies in this field increasingly implement hypothesis-driven methodologies, as they would enhance the interpretability, replicability, and theoretical development of embodied cognitive neuroscience in real-world contexts.

Nevertheless, among the principal results of this review, it is possible to identify a significant interest in the relationship between different variables of the indoor built environment and psycho-physiological states, although studies of isolated variables proliferate with little holistic vision of multi-modal phenomena and cross-modal stimuli interactions. For instance, thermal comfort was the most commonly investigated IEQ variable in relation to heart activity and skin temperature. These findings are relevant considering the impact of the environment on the health–disease continuum and everyday life, as the opportunity that they offer to generate strategies for interventions and improving quality of life is significant. Our research also highlights the need to shift toward using advanced technologies, such as Mobile Brain/Body Imaging (MoBI), to capture real-time biometric data in natural settings and appropriate theoretical paradigms such as the 3E-cognition perspective to interpret and guide the results. These technologies can be extrapolated and applied to everyday scenarios, including health-related, work, and educational contexts.

## Figures and Tables

**Figure 1 ijerph-22-00824-f001:**
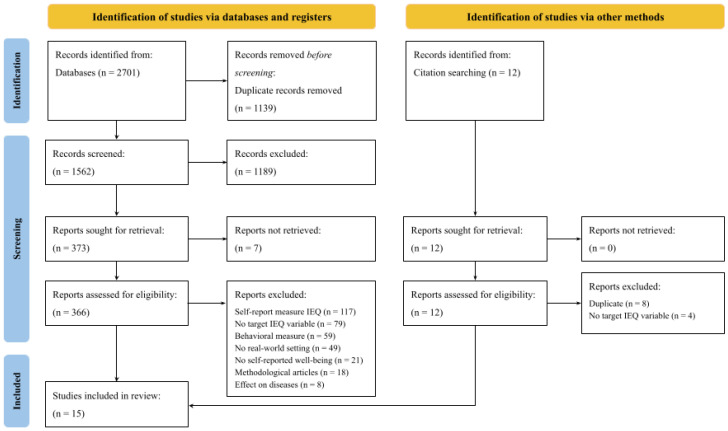
PRISMA flow diagram.

**Figure 2 ijerph-22-00824-f002:**
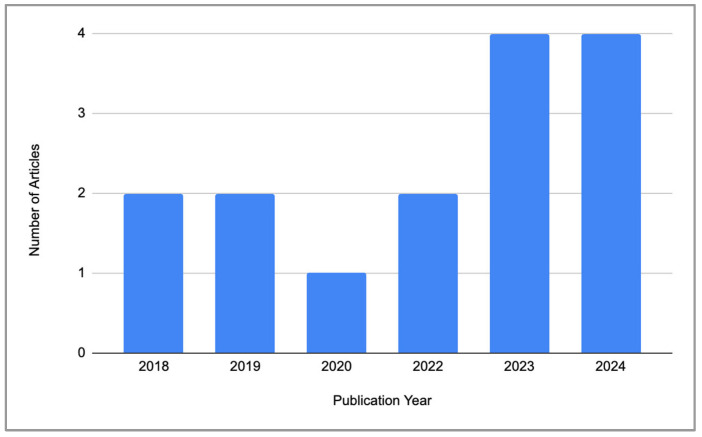
Distribution of included articles by year of publication.

**Figure 3 ijerph-22-00824-f003:**
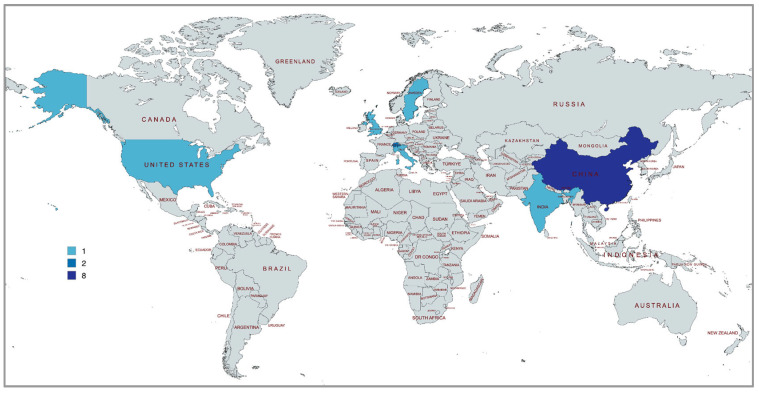
Geographical distribution of the included articles.

**Table 1 ijerph-22-00824-t001:** Classification of physiological measures according to their relationship with cognitive processes and their degree of directness. Measures range from direct neural correlates to indirect autonomic indices, all contributing to an embodied, environmentally scaffolded, and enactive understanding of human cognition as outlined in the 3E-Cognition framework.

Physiological Measure	Relationship to Cognitive Processes	Degree of Directness	Notes
EEG (Electroencephalography)	Direct measure of neural activity underpinning cognition	Direct	Gold standard in Mobile Brain/Body Imaging (MoBI).
ECG (Electrocardiography)	Indirect measure via cardiac–brain interactions (e.g., HEP)	Moderately Direct	Useful for tracking interoceptive processes.
HEP (Heartbeat-Evoked Potential)	Direct neural correlate of interoceptive awareness and bodily self-consciousness	Direct	Emerging method with clear links to embodied cognition.
Heart Rate (HR)	Reflects autonomic nervous system regulation linked to stress, attention, and emotion	Indirect	Important for embodied cognition and well-being.
Heart Rate Variability (HRV)	Indicates cognitive–emotional regulation via autonomic function	Moderately Direct	Well-established proxy for emotion–cognition coupling.
Electrodermal Activity (EDA)	Reflects arousal, an emotional response, and the cognitive load	Indirect	Sensitive to attentional and affective dynamics.
Skin Temperature	Proxy for emotional and autonomic state changes	Indirect	Useful to infer comfort and stress but less specific.
Eye Tracking	Indicated attentional focus and cognitive strategies	Direct	Especially valuable when combined with EEG in MoBI.

**Table 2 ijerph-22-00824-t002:** PICO framework for inclusion and exclusion criteria.

PICO	Inclusion Criteria	Exclusion Criteria
Population	Neurotypical human participants	None
Intervention	Exposure to at least one of the four main factors of indoor environmental quality—air quality, thermal comfort, noise, or lighting—in real-world scenarios instead of traditional laboratory setups	None
Comparators	Not applicable	Not applicable
Outcomes	Data related to the acquisition of physiological variables and self-reported measures of well-being	None
Other	Empirical design, published in peer-reviewed journals, and available in English or Spanish from any geographic region with no time restrictions.	None

**Table 3 ijerph-22-00824-t003:** Summary of the risk of bias.

Case Series Studies										
Reference	Q1	Q2	Q3	Q4	Q5	Q6	Q7	Q8	Q9	Q10
Wang et al., 2018 [108]	Yes	Yes	Yes	Unclear	Yes	Yes	NA	Yes	Yes	Yes
Chinazzo et al., 2018 [101]	Yes	Yes	Yes	Unclear	Yes	Yes	NA	Yes	Yes	Yes
Snow et al., 2019 [107]	Yes	Yes	Yes	Unclear	Yes	Yes	NA	Yes	Yes	Yes
Chinazzo et al., 2019 [102]	Yes	Yes	Yes	Unclear	Yes	Yes	NA	Yes	Yes	Yes
Song et al., 2020 [97]	Yes	Yes	Yes	Unclear	Yes	Yes	NA	Yes	Yes	Yes
Barbic et al., 2022 [103]	Yes	Yes	Yes	Unclear	Yes	Yes	NA	Yes	Yes	Yes
Wang et al., 2023 [98]	Yes	Yes	Yes	Unclear	Yes	Yes	NA	Yes	Yes	Yes
Zhou et al., 2023 [100]	Yes	Yes	Yes	Unclear	Yes	Yes	NA	Yes	Yes	Yes
Gao et al., 2023 [95]	Yes	Yes	Yes	Unclear	Yes	Yes	NA	Yes	Yes	Yes
Roy et al., 2024 [106]	Yes	Yes	Yes	Unclear	Yes	Yes	NA	Yes	Yes	Yes
Fischl & Johansson, 2024 [105]	Yes	Yes	Yes	Unclear	Yes	Yes	NA	Yes	Yes	Yes
Fanpu et al., 2024 [94]	Yes	Yes	Yes	Unclear	Yes	Yes	NA	Yes	Yes	Yes
Beaudette et al., 2024 [104]	Yes	Yes	Yes	Unclear	Yes	Yes	NA	Yes	Yes	Yes
**Cross-Sectional Studies**										
**Reference**	**Q1**	**Q2**	**Q3**	**Q4**	**Q5**	**Q6**	**Q7**	**Q8**		
Hu et al., 2022 [96]	Yes	Yes	Yes	Yes	Unclear	Unclear	Yes	Yes		
Wu & Wagner, 2023 [99]	Yes	Yes	Yes	Yes	Unclear	Unclear	Yes	Yes		

Case series studies: Q1: Were there clear criteria for inclusion in the case series? Q2: Was the condition measured in a standard, reliable way for all participants included in the case series? Q3: Were valid methods used for the identification of the condition for all participants included in the case series? Q4: Did the case series have consecutive inclusion of participants? Q5: Did the case series have complete inclusion of participants? Q6: Was there clear reporting of the demographics of the participants in the study? Q7: Was there clear reporting of clinical information of the participants? Q8: Were the outcomes or follow-up results of cases clearly reported? Q9: Was there clear reporting of the presenting site(s)/clinic(s) demographic information? Q10: Was statistical analysis appropriate? Cross-Sectional Studies: Q1: Were the criteria for inclusion in the sample clearly defined? Q2: Were the study subjects and the setting described in detail? Q3: Was the exposure measured in a valid and reliable way? Q4: Were objective, standard criteria used for the measurement of the condition? Q5: Were confounding factors identified?; Q6: Were strategies to deal with confounding factors stated? Q7: Were the outcomes measured in a valid and reliable way? Q8: Was appropriate statistical analysis used? NA: not applicable.

**Table 4 ijerph-22-00824-t004:** Main objectives and IEQ variables studied in the included articles.

Reference	Aim	Study Type	IEQ Variable	Sample Size
Chinazzo et al., 2018 [101]	To investigate the effect of daylight transmitted through three colored glazing types on thermal responses and overall comfort at three temperature levels.	Experimental	Thermal comfort	75
Wang et al., 2018 [108]	To study the effect of the indoor environment on students’ learning performance in summer.	Experimental	Thermal comfort	12
Chinazzo et al., 2019 [102]	To investigate the influence of daylight on thermal responses, intended as both subjective thermal perceptions and physiological responses, by studying different combinations of daylight and temperature levels.	Experimental	Lighting and thermal comfort	84
Snow et al., 2019 [107]	To measure changes in human performance, physiological, neurophysiological (EEG), and psychological factors as a result of elevated indoor CO2 concentrations and to validate EEG as an objective measurement of sleepiness for studies concerned with the effect of the indoor environment on humans.	Experimental	Air quality	31
Song et al., 2020 [97]	To identify the important local thermal conditions for sleeping thermal comfort.	Experimental	Thermal comfort	12
Barbic et al., 2022 [103]	To evaluate the effects of reducing classroom temperatures on cognitive performance and to evaluate the associated changes in cardiac autonomic control in a class of undergraduate students.	Experimental	Thermal comfort and air quality	15
Hu et al., 2022 [96]	To analyze gender differences in thermal comfort, work performance, and Sick Building Syndrome symptoms and to define optimal temperature ranges while considering gender differences in the three above-mentioned aspects in classrooms in winter.	Observational	Thermal comfort	2110
Gao et al., 2023 [95]	To analyze the dynamic changes and effects of indoor temperature and exercise behavior on human thermal comfort and establish an evaluation mechanism of exercise thermal comfort.	Observational	Thermal comfort	45
Wang et al., 2023 [98]	To provide theoretical foundations for developing healthy building smell-scapes with essential oils.	Experimental	Thermal comfort, lighting, and air quality	10
Wu & Wagner, 2023 [99]	To examine the effect of outdoor short-term thermal history on the thermal comfort and physiological responses of humans in the indoor environment.	Observational	Thermal comfort	32; 345
Zhou et al., 2023 [100]	To determine the preferred indoor air velocity under hot and humid conditions and to examine the effects of elevated air movement on subjective perceptions and skin temperature.	Experimental	Thermal comfort	36
Beaudette et al., 2024 [104]	To investigate the use of readily available heating devices in typical indoor climate-controlled environments, contexts in which some users may feel uncomfortably cool or cold.	Experimental	Thermal comfort	17
Fanpu et al., 2024 [94]	To explore the best artificial illumination supplement under the comprehensive evaluation of vision, emotion, and cognition in different periods.	Experimental	Lighting	30
Fischl & Johansson, 2024 [105]	To integrate digital occupancy assessment methods to understand indoor lighting conditions on occupant well-being.	Experimental	Lighting	13
Roy et al., 2024 [106]	To explore the influence of three different lighting conditions on participants’ perceptions of task lighting attributes, aspects of room esthetics, and their physiological responses.	Experimental	Lighting	24

**Table 5 ijerph-22-00824-t005:** Task performed and research settings.

Reference	Task	Setting
Chinazzo et al., 2018 [101]	Office tasks (paper-based performance tests such as the d2 test)	Office-like room
Wang et al., 2018 [108]	Learning performance test	Experimental room in rural primary and secondary school
Chinazzo et al., 2019 [102]	Office tasks (assigned paper-based)	Office-like room
Snow et al., 2019 [107]	Cognitive test (Stroop test, shifting attention task, continuous performance test, four-part continuous performance test)	Office
Song et al., 2020 [97]	Sleeping behavior	Residential house
Barbic et al., 2022 [103]	Cambridge Brain Science Task	University classroom
Hu et al., 2022 [96]	Course-related activities such as reading books or completing homework	University classroom
Gao et al., 2023 [95]	Field test lasting >100 min (preparation, warm-up, and testing stage).	Physical fitness training center
Wang et al., 2023 [98]	Cognitive tasks/tests (Stroop, N-back, visual search task, and digital memory span test)	Office
Wu & Wagner, 2023 [99]	No specific task (indoor staying)	University dormitory rooms
Zhou et al., 2023 [100]	Office tasks (computer work or reading)	Office-like rooms
Beaudette et al., 2024 [104]	Office tasks (self-directed desk work)	Office-like room
Fanpu et al., 2024 [94]	Reading tasks (Anfimov letter recognition table, number proofreading table, and Randall ring checklist)	Reading space of the university library
Fischl & Johansson, 2024 [105]	Simple (navigation map) and complex (floor plans and facades of their homes) drawing tasks	Students group rooms
Roy et al., 2024 [106]	Reading (common textbook) and writing (copy diagrams and paragraphs) tasks	Tertiary educational institute classroom

**Table 6 ijerph-22-00824-t006:** Instruments used and measurements conducted.

Reference	Data Collection Technique	Main Biomarker	Self-Report	Cross-Test Between Variables
Chinazzo et al., 2018 [101]	Surface thermometer; wristband sensor	Skin temperature; heart rate; skin conductance	Thermal comfort (based on a 7-point scale of the ASHRAE); ergonomics of the thermal environment	No
Wang et al., 2018 [108]	Electronic sphygmomanometer; electronic thermometer	Blood pressure; heart rate; body temperature	ASHRAE 7-point thermal sensation scale	Yes
Chinazzo et al., 2019 [102]	Surface thermometer	Skin temperature	Ergonomics of the thermal environment	No
Snow et al., 2019 [107]	EEG; Electrodes; Finger clip; Abdominal belt	Heart rate; Respiration rate; Skin temperature; EEG frequency bands	Sick Building Syndrome symptoms; Stanford Sleepiness Scale; Positive and Negative Affective State	Yes
Song et al., 2020 [97]	**Surface thermometer**	Skin temperature	Thermal sensation, comfort, and acceptability evaluation scales (TSV; TCV; TAV)	Yes
Barbic et al., 2022 [103]	ECG	Heart rate; respiration rate	Questionnaire for the thermal comfort survey (modified by Wang)	No
Hu et al., 2022 [96]	Thermocouple; fingertip pulse oximeter	Skin temperature; blood oxygen saturation; heart rate	Thermal comfort; self-estimated work performance; Sick Building Syndrome symptoms (based on a 7-point scale of the ASHRAE)	No
Gao et al., 2023 [95]	Infrared imager; four-channel thermometer; chest strap sensor; blood pressure monitor	Skin temperature; heart rate; blood pressure	TSV; TCV; TPV; TSL; FSV	No
Wang et al., 2023 [98]	Holter monitor; sphygmomanometer	Heart rate; heart rate variability; blood pressure	POMS questionnaire; indoor air quality acceptability scale; odor scale	No
Wu & Wagner, 2023 [99]	Surface thermometer; blood pressure monitor	Skin temperature; heart rate; blood pressure	Thermal sensation, preference, comfort, and acceptability (based on a 7-point scale of the ASHRAE)	Yes
Zhou et al., 2023 [100]	Surface thermometer	Skin temperature	Scales for perceptions on temperature, humidity, and air movement (TSV; HSV; AMV; TCV; TS; HS; AS; TAV; HAV; AMA; TPV; HPV; AVP)	Yes
Beaudette et al., 2024 [104]	Thermistors	Skin temperature	Adapted ASHRAE scale (perceived thermal sensation and comfort)	Yes
Fanpu et al., 2024 [94]	Heart rate band	Heart rate variability	Positive and Negative Emotion Scale; visual discomfort scale; office lighting survey scale; Subjective alertness scale	No
Fischl & Johansson, 2024 [105]	Galvanic skin response sensor; wristband sensor	Skin conductance; heart rate	7-point Likert scale (satisfaction, pleasantness, novelty, and other perceptions); Self-Assessment Manikin (emotional reaction)	No
Roy et al., 2024 [106]	Digital sphygmomanometer; fingertip pulse oximeter	Blood pressure; heart rate	5-point semantic differential scale (aspects of lighting of tasks and room esthetics)	No

TSV: thermal sensation vote; TCV: thermal comfort vote; TAV: thermal acceptability vote; TPV: thermal preference vote; TSL, thermal satisfaction level; FSV, fatigue sensation vote; POMS: profile of mood states; HSV: humidity sensation vote; AMV: air movement sensation vote; TS: thermal satisfaction; HS: humidity satisfaction; AS: air movement satisfaction; HAV: humidity acceptability vote; AMA: air movement acceptability vote; HPV: humidity preference vote; AVP: air movement preference vote.

## Data Availability

The dataset is available at the Open Science Framework (OSF): https://osf.io/qc5tx/.

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
