# Peer review of "Neuroscientific Insights into the Built Environment: A Systematic Review of Empirical Research on Indoor Environmental Quality, Physiological Dynamics, and Psychological Well-Being in Real-Life Contexts"

_ijerph, 2025, doi:10.3390/ijerph22060824_

Round 1

Reviewer 1 Report

Comments and Suggestions for Authors

In the systematic review study, neuroscientific studies on indoor comfort conditions are evaluated. Some deficiencies listed below were identified in the study:

  • The introductory section commences with a discussion of urbanization; however, the focus of the study pertains to the built environment, rather than urbanization. Consequently, the initial paragraph should be revised to align with the study's objective.
  • The subsequent section, titled "The Impact of the Built Environment on Health," presents numerous case studies. However, a unifying theme that links these studies is lacking. To enhance the clarity and coherence of the text, this section would benefit from a more structured approach.
  • The study's authors did not clarify which keywords were used or how they were entered into the electronic databases. These details must be provided.
  • Additionally, the methodology employed to delineate the year limit of the publications included in the study merits clarification. It is imperative to ascertain the rationale behind this criterion. The rationale behind the selection of only the last seven years is also unclear.
  • The inquiries employed in the JBI Critical Appraisal Tool must also be specified. Furthermore, the appropriateness of these questions for the review is to be determined. It is imperative that these questions be articulated in a clear and precise manner to ensure the integrity of the study's findings.

Author Response

Reviewer 1:

  1. The introductory section commences with a discussion of urbanization; however, the focus of the study pertains to the built environment, rather than urbanization. Consequently, the initial paragraph should be revised to align with the study's objective.

DONE. We thank the reviewer for this helpful observation. The introduction has been revised to focus directly on the built indoor environment and its relevance to human experience, ensuring alignment with the study’s objective. The updated paragraph (Line 43) is as follows:

“The built environment can be defined as the natural environment modified by human conceptualizations and actions (Bartuska, 2007; Portella, 2014). Considering the peculiarities of modern life, people spend approximately 90% of their time indoors (Azzazy et al., 2021; Klepeis et al., 2001; Schweizer et al., 2007), which makes understanding the relationship between indoor environmental characteristics and human functioning especially relevant. Researchers have explored several components of the built environment (e.g., light exposure, acoustic conditions, temperature, and air quality) to examine its relationship with physiology (e.g., cortisol levels, circadian rhythm; (Beil & Hanes, 2013; Stevens & Rea, 2001), psychological states (e.g., happiness, irritability, stress; Sullivan & Chang, 2011; Zainal & Hosni, 2022), and cognitive processes (e.g., attention, learning, memory; Besser et al., 2018; Keis et al., 2014; Marchand et al., 2014; Möystad, 2017). Findings show that some features of the environment can promote or disturb mental health and cognitive functioning (Karakas & Yildiz, 2020; Moore et al., 2018; Rhodes et al., 2018), and can have a direct impact on neurobiology (Djebbara et al., 2022).”

  1. The subsequent section, titled "The Impact of the Built Environment on Health," presents numerous case studies. However, a unifying theme that links these studies is lacking. To enhance the clarity and coherence of the text, this section would benefit from a more structured approach.

DONE. We thank the reviewer for this constructive comment. In response, we have restructured the section to provide a clearer and more coherent narrative linking the cited studies. The revised version introduces a conceptual progression from negative to positive effects of indoor environmental conditions, followed by a synthesis under the framework of Indoor Environmental Quality (IEQ). This structure enhances the thematic unity of the section and aligns it more closely with the review’s core focus. The updated section (Line 59) now reads: 

“The built indoor environment can influence mental health and well-being through both negative and positive environmental features. Several studies have shown that certain stressors commonly found in indoor settings—such as crowding, noise, and environments perceived as unsafe—may negatively affect psychological states, leading to chronic stress and symptoms associated with psychiatric conditions (Evans et al., 2003; Lederbogen et al., 2011, 2013; Matthews & Yang, 2010). 

In contrast, other characteristics associated with indoor spaces or their immediate surroundings have been shown to foster psychological well-being. For example, having green views from indoors—such as visual access to trees or vegetation through windows—has been linked to increased perceived well-being and reduced mental fatigue (Day, 2008; Sullivan & Chang, 2011; Yuen & Nyuk Hien, 2005). Similarly, exposure to natural light within indoor spaces may alleviate symptoms of seasonal depression (Elliott et al., 1993; Matthews & Yang, 2010; Taylor et al., 1997). Overall, improvements in the indoor environment have been associated with reduced physiological and psychological distress in various contexts, including workplaces and healthcare settings (Beil & Hanes, 2013; Beukeboom et al., 2012; Codinhoto et al., 2009; Lottrup et al., 2013; Ulrich, 1981; van den Berg et al., 2010; Ward Thompson et al., 2012).

The framework commonly used to conceptualize these effects is Indoor Environmental Quality (IEQ), which refers to the set of physical conditions in indoor spaces that affect occupants' health, comfort, and well-being (Abdulaali et al., 2020; Mallawaarachchi et al., 2012; Rohde et al., 2020; Steinemann et al., 2017). According to systematic reviews, IEQ dimensions such as air quality, thermal comfort, acoustic conditions, and lighting are considered positive when they help reduce discomfort or promote cognitive and emotional functioning (Al horr et al., 2016; Salonen et al., 2013). These factors have been studied in various contexts. For example, Turunen and colleagues (2014) found associations between classroom noise and poor ventilation and symptoms like fatigue and headaches in students. Similarly, Salonen and colleagues (2013) showed that adequate ventilation and favorable visual conditions—such as appropriate lighting levels—can positively affect the well-being of individuals in healthcare environments.”

  1. The study's authors did not clarify which keywords were used or how they were entered into the electronic databases. These details must be provided.

DONE. We appreciate the reviewer’s useful observation. We have revised the Methods section to clarify the search strategy. We now specify all keywords used, provide an example of a search string, and note that the full list of search queries is available in the study protocol (updated). The revised paragraph now reads (Line 241):

“A search strategy was developed following the Peer Review of Electronic Search Strategies (PRESS) checklist (McGowan et al., 2016). The search strategy was adapted to each database. The search process was conducted by two members of the research team (AGC and MAP) after a preliminary, non-systematic review of keywords related to indoor environmental quality. The search strategy aimed to identify studies exploring the relationship between indoor environmental quality (IEQ) and physiological or neurophysiological responses. To that end, each IEQ-related term—“indoor thermal comfort”, “indoor air quality”, “indoor noise”, “indoor light”, and “indoor environment”—was combined individually with each physiological keyword: “physiolog*”, “heart*”, “brain”, “respiration”, and “skin temperature”. For example, one of the search strings was: "indoor air quality" AND "physiolog*". A full list of search strings is available in the study protocol. Keywords were searched in the articles’ abstracts. Systematics reviews and conference abstracts were excluded from the search.”

  1. Additionally, the methodology employed to delineate the year limit of the publications included in the study merits clarification. It is imperative to ascertain the rationale behind this criterion. The rationale behind the selection of only the last seven years is also unclear.

ADDRESSED. We thank the reviewer for this observation and appreciate the opportunity to clarify. No time limit was applied as part of the inclusion criteria. The fact that all selected studies were published within the last seven years was not a result of a predefined restriction, but rather an outcome of the search and selection process. We have reinforced this point in both the Methods and Results sections to avoid further misunderstanding. Clarifications added (Line 222 and 300):

Methods section: “All available publications in English or Spanish were eligible for inclusion, and no temporal limits (i.e., publication year) were applied at any stage of the search or selection process.”

Results section: “Although no time restriction was established a priori, all included articles were published between 2018 and 2024.”

  1. The inquiries employed in the JBI Critical Appraisal Tool must also be specified. Furthermore, the appropriateness of these questions for the review is to be determined. It is imperative that these questions be articulated in a clear and precise manner to ensure the integrity of the study's findings.

DONE. We appreciate the reviewer pointing out that the items for evaluation were missing. This was a problem with the import-export of the document. The full description of the items is available as a footnote on the table (Line 326).

Case Series Studies: Q1 Were there clear criteria for inclusion in the case series?; Q2 Was the condition measured in a standard, reliable way for all participants included in the case series?; Q3 Were valid methods used for identification of the condition for all participants included in the case series?; Q4 Did the case series have consecutive inclusion of participants?; Q5 Did the case series have complete inclusion of participants?; Q6 Was there clear reporting of the demographics of the participants in the study?; Q7 Was there clear reporting of clinical information of the participants?; Q8 Were the outcomes or follow up results of cases clearly reported?; Q9 Was there clear reporting of the presenting site(s)/clinic(s) demographic information?; Q10 Was statistical analysis appropriate?

Cross-Sectional Studies: Q1 Were the criteria for inclusion in the sample clearly defined?; Q2 Were the study subjects and the setting described in detail?; Q3 Was the exposure measured in a valid and reliable way?; Q4 Were objective, standard criteria used for measurement of the condition?; Q5 Were confounding factors identified?; Q6 Were strategies to deal with confounding factors stated?; Q7 Were the outcomes measured in a valid and reliable way?; Q8 Was appropriate statistical analysis used?

NA: not applicable

Reviewer 2 Report

Comments and Suggestions for Authors

The authors present results of a systematic review regarding methodological aspects related to the assessment of indoor environmental quality.
Given the extended period of time, humans spend indoors and known influences of the IEQ on human health and well-being, the topic is worth being investigated. At the same time, methods applied in the field are varying, so that their review could help bringing order to the field and improve research methods in the future. Hence, the topic of this manuscript is well chosen and relevant.
However, there are two main limitations. First, the search strategy is not reproducible as the chosen search terms and phrases are not reported. Hence, I cannot judge the suitability of these and their implications for the review process. A final assessment of the quality of the methodology is therefore not possible. Second, the presentation of results in its current form remains narrative and vague. Authors summarize the methods used of a limited number of articles included in their review (N=16), but do not get into details or draw in-depth conclusions. The suggestions made regarding the increased use of HEP and MoBI appear to be motivated by personal interests rather than resulting from the analysis of the reviewed papers as further detailed below. At least, in its current form of the manuscript, there appears to be a larger argumentative gap between the results presented and the recommendations made. Hence, according to this reviewer, rethinking the approach to review and interpretation is highly recommended within a major revision.
Specific comments and arguments for above summary are as follows. Numbers mark the first line of the manuscript, the comment refers to.
42 and follows. This first paragraph is a bit confusing and readers will be not know the focus of the article. The reason is that the terms built environment, urban environment and natural environment are mixed and a strong focus in the indoor environment – the topic of this manuscript – is diffuse. I suggest outlining from the beginning of the paragraph, where the journey is heading for. Generally, the first paragraph could be omitted completely as it does not add much to the manuscript. Starting with the second paragraph would be fine.
54. Please be aware that the figure of 90% include periods in cars and other transport means, typically not considered part of the built environment – only of indoors.
70. It is not clear, why traffic-related pollution is highlighted here in a review of INDOOR environmental quality.
74. Same as 70, green areas are not indoors, while some plants are. The following part regarding view from indoors to green spaces outdoors is relevant though.
91. I appreciate that the authors also mention positive effects of IEQ conditions. At the same time, this sentence does not clarify what these are. Air quality, ergonomics and other conditions mentioned here, are not necessarily positive. What characteristics makes them positive? Please elaborate further. In addition, why is ergonomics included here as IEQ condition, but not dealt with in the later review?
117. The concept of 3E-Cognition appears to be central for the review – at least it is further mentioned in line 172. Explanations of this concept are not existing and should be added. It is also not clear, how the sentences in lines 109 to 117 relate to 3E-Cognition. I suggest to clarify these points, for example by clearing including the three key terms emergent, embodies, and extended.
125. This paragraph appears an unsystematic summary of some findings related to thermal aspects. How have those examples been selected and why are other IEQ-components ignored here?
142 and in general throughout the paper. A clear definition of neuroscience research and components would be helpful. Without such clarification, readers will be confused, why measures like heart rate, EDA, skin temperature are included in this review, considering that they rather indirectly inform about cognitive processes in comparison to ECG and others. Adding a classification of the relationship between such measures and cognitive processes as well as the degree, such measures inform about cognitive processes would be crucial for readers to follow the authors line of argument.
177. Similar to 142, it would be helpful to define the authors’ interpretation of well-being and how it related to aspects like thermal sensation and heart rate.
194. (major) The limitation to studies that include both measures, i.e. physiological and self-reported appears a strong limitation of the approach and tremendously reducing the number of included articles. At the same time, review results presented in no way make comparisons between these two types of measures. Hence it is questionable that such inclusion criteria is justified as results would gain much more when based on a larger overview of studies in the field of IEQ and human reactions (both physiological and self-report).
206. Avoid writing in verbal language style, i.e. do not use haven’t and similar phrases but write have not
216. (major) The search strategy is poorly described and the protocol on osf does not give further information. For a reproducible description, the search terms and phrases, i.e. their combination with for example “AND” or “OR” is crucial. Without such information, the method is not reproducible and the quality of the review process cannot be judged.
255. (major) The resulting number of included articles is very small, which – as mentioned above – might be due to a too restricted inclusion criteria or the search terms used. Such small number of articles can be justified when an in-depth analysis of their methods would be conducted. For example, going in-depth into the instruments used, their similarities and differences, which starts from scales used. In its current form, results remain superficial and narrative and do not add much value to existing literature.
295. Table 2. Please add a footnote explaining at least the topics of Q1 to Q10, so that readers can understand the information in the table without having to look at other literature sources.
309 and other lines. Lightning is something completely different than lighting.
336. (major) The definition of “real-life context”, which appears a major inclusion criteria, is not clear. It appears that many studies selected are not performed in classic laboratories, which is as targeted, while they are done in specially prepared rooms not used for daily activities. For example, this applies to Chinazzo et al which took place not in a commonly used office room, but only an office-like room specially prepared for the experiment. Similar things can be said about Zhou et al. Also Wang et al is described as “experimental room”. As such, participants of those studies are not in their normal work environment, but in a still artificial one. It would be crucial to extend the description of the settings in order to clarify the “naturalness” of these settings. Up to which point is a lab a lab? When, i.e starting from which properties, does it turn into a real-life context? What are the influences of such properties on the (ecological) validity of results? These are crucial questions that could increase the value of this review beyond the current narrative status.
355. Why are iButtons in a different category from surface thermometers? iButtons are a brand name – truly commonly used – but at the same time, they measure surface temperature and are hence surface thermometers and not a category on their own.
Results in general. (major) The results section remains narrative and largely superficial. There are many more aspects related to methodological aspects which would be interesting to compare. For example, what are the experimental approaches of these studies? Are they rather observational or do they manipulate specific IEQ parameters systematically and experimentally? What is the range of conditions? What is the size of samples? And so on. Right now, the results focus on a very limited number of aspects related to complex methodological questions and has not much value.
409 and complete paragraph. (major) This part of the discussion is highly speculative. Authors appear to credit the COVID-19 pandemic leading to an increase in studies. There could be numerous other reasons for the increase, for example the general increase in publications in all fields over the last years. Hence, their argument appears not justified and needs to be made on more solid arguments and facts. For example, compare the observed increase of included studies compared to general publication trends. Also consider technological developments, which ease such studies and hence increase also the output of these. The current line of argument is too weak to be justified and ignore potential other explanations. 
425. The share of studies related to location should be set in relation to overall publication trends. Chinese publications have a large general share in worldwide scientific publications in all fields.
445. Due to the lack of information regarding the search terms, it is not clear whether the lack of studies on noise is true or just a result of the search strategy.
481. When asking for the meaning of well-being, I would expect authors to reflect more on work reflecting the definition of well-being. Right now, Altomonte et al is mentioned, but there are other recent publications dealing with definitions. Why was Altomonte highlighted here and not others?
517. The diversity of hypotheses is not part of the results section. Please consider changing to “aims”, which have been briefly touched, here or add a summary of hypotheses to the results section.
533. It is not clear, why authors highlight HEP. Such recommendations would require much more in-depth analysis and explanation of its advantages over other methods, which at the same time should include its limitations.
539 and follows. This part of the discussion appears to be disconnected from the results. Hypotheses have not been analyzed in the reviewed articles. It would be crucial to align results and discussion, hence an in-depth analysis of the current status of hypothesis-driven analysis strategies would be needed before it can be discussed here.
562. “The importance … is increasing” This sentence and several similar statements by the authors appear to be based on their own opinion. No reference is given here to justify such statements hence it remains unknown, how authors got to such statements and the agenda behind it.
598. (major) Similar to 533, it is not clear, why MoBI is highlighted and recommended. Such recommendation is not a consequence of the results and therefore needs much more argumentation and discussion of its pros and cons compared to other approaches. Without such line of argument, it could be just the authors’ own interest.
607. Small typo By
638. (major) Authors state here, that “it is possible to identify a significant relationship between different variables …” However, their review of the included articles does not include the results sections of those articles, solely the methodological aspects. Hence, this conclusion is not based on the results unless such analysis is added to the results section. The same statement is included in the abstract and not justified there neither.
642. Similar to 638, the degree of influence of IEQ on any parameter like heat rate and skin temperature is not part of the results section of this review. The results section solely looked at the number of studies that investigated such influence. 
645. As mentioned above, the justification of the recommendation for MoBI is not convincing in the discussion section and needs to be improved.

Author Response

Reviewer 2:

  1. First, the search strategy is not reproducible as the chosen search terms and phrases are not reported. Hence, I cannot judge the suitability of these and their implications for the review process. A final assessment of the quality of the methodology is therefore not possible.

DONE. We appreciate the reviewer’s critical observation. To ensure reproducibility and transparency, we have revised the Methods section to report the full list of keywords used, and clarify how the search terms were combined using Boolean logic. We have also included an example search string and noted that the complete list of queries is available in the study protocol (updated). The revised paragraph now reads (Line 241):

“A search strategy was developed following the Peer Review of Electronic Search Strategies (PRESS) checklist (McGowan et al., 2016). The search strategy was adapted to each database. The search process was conducted by two members of the research team (AGC and MAP) after a preliminary, non-systematic review of keywords related to indoor environmental quality. The search strategy aimed to identify studies exploring the relationship between indoor environmental quality (IEQ) and physiological or neurophysiological responses. To that end, each IEQ-related term—“indoor thermal comfort”, “indoor air quality”, “indoor noise”, “indoor light”, and “indoor environment”—was combined individually with each physiological keyword: “physiolog*”, “heart*”, “brain”, “respiration”, and “skin temperature”. For example, one of the search strings was: "indoor air quality" AND "physiolog*". A full list of search strings is available in the study protocol. Keywords were searched in the articles’ abstracts. Systematics reviews and conference abstracts were excluded from the search.”

  1. Second, the presentation of results in its current form remains narrative and vague. Authors summarize the methods used of a limited number of articles included in their review (N=16), but do not get into details or draw in-depth conclusions. The suggestions made regarding the increased use of HEP and MoBI appear to be motivated by personal interests rather than resulting from the analysis of the reviewed papers as further detailed below.

At least, in its current form of the manuscript, there appears to be a larger argumentative gap between the results presented and the recommendations made. Hence, according to this reviewer, rethinking the approach to review and interpretation is highly recommended within a major revision.

DONE/Discussed. We appreciate the reviewer’s insightful comment regarding the results presentation. As explained in the Methods section ("Synthesis of Results"), we selected a narrative approach to synthesize our findings. This choice was based on the substantial heterogeneity across the included studies regarding (1) experimental designs, (2) types of variables measured, and (3) reported outcomes. Given this variability, a meta-analysis—the gold standard for quantitative synthesis—was not feasible. We have now clarified this rationale more explicitly in the Methods section. The revised text reads (Line 276):

"Following the narrative methodology proposed by Arksey and O'Malley (2005), data were summarized according to extraction categories aimed at answering the review's objectives and questions. Data are presented graphically and in tables. Due to the high heterogeneity of study aims, experimental designs, and measured outcomes, a meta-analysis was not performed (Field et al., 2010; Soni et al., 2023).”

  1. 42 and follows. This first paragraph is a bit confusing and readers will be not know the focus of the article. The reason is that the terms built environment, urban environment and natural environment are mixed and a strong focus in the indoor environment – the topic of this manuscript – is diffuse. I suggest outlining from the beginning of the paragraph, where the journey is heading for. Generally, the first paragraph could be omitted completely as it does not add much to the manuscript. Starting with the second paragraph would be fine.

DONE. We appreciate the reviewer’s insightful feedback. In response, we have rewritten the introductory paragraph to avoid conceptual overlap between the built, urban, and natural environments and to emphasize the manuscript’s focus on the indoor built environment. The revised version provides a more direct entry into the topic, enhancing clarity and alignment with the article’s objectives. The revised paragraph (Line 43) now reads:

“The built environment can be defined as the natural environment modified by human conceptualizations and actions (Bartuska, 2007; Portella, 2014). Considering the peculiarities of modern life, people spend approximately 90% of their time indoors (Azzazy et al., 2021; Klepeis et al., 2001; Schweizer et al., 2007), which makes understanding the relationship between indoor environmental characteristics and human functioning especially relevant. Researchers have explored several components of the built environment (e.g., light exposure, acoustic conditions, temperature, and air quality) to examine its relationship with physiology (e.g., cortisol levels, circadian rhythm; (Beil & Hanes, 2013; Stevens & Rea, 2001), psychological states (e.g., happiness, irritability, stress; Sullivan & Chang, 2011; Zainal & Hosni, 2022), and cognitive processes (e.g., attention, learning, memory; Besser et al., 2018; Keis et al., 2014; Marchand et al., 2014; Möystad, 2017). Findings show that some features of the environment can promote or disturb mental health and cognitive functioning (Karakas & Yildiz, 2020; Moore et al., 2018; Rhodes et al., 2018), and can have a direct impact on neurobiology (Djebbara et al., 2022).”

  1. 54. Please be aware that the figure of 90% include periods in cars and other transport means, typically not considered part of the built environment – only of indoors.

DONE. We thank the reviewer for this important clarification. The sentence has been revised to more accurately reflect the source data by referring to “time spent indoors” rather than “in the built environment.” This adjustment improves conceptual clarity and ensures a more accurate representation of the original data. The revised sentence (Line 44) now reads:

“Considering the peculiarities of modern life, people spend approximately 90% of their time indoors (Azzazy et al., 2021; Klepeis et al., 2001; Schweizer et al., 2007).”

  1. 70. It is not clear, why traffic-related pollution is highlighted here in a review of INDOOR environmental quality.

DONE. We thank the reviewer for this thoughtful observation. In order to maintain a clear focus on indoor environmental quality (IEQ), we have removed the example related to traffic-related pollution. The revised paragraph now emphasizes environmental stressors more directly associated with indoor settings. The updated paragraph (Line 59) now reads:

“The built indoor environment can influence mental health and well-being through both negative and positive environmental features. Several studies have shown that certain stressors commonly found in indoor settings—such as crowding, noise, and environments perceived as unsafe—may negatively affect psychological states, leading to chronic stress and symptoms associated with psychiatric conditions (Evans et al., 2003; Lederbogen et al., 2011, 2013; Matthews & Yang, 2010).”

  1. 74. Same as 70, green areas are not indoors, while some plants are. The following part regarding view from indoors to green spaces outdoors is relevant though.

DONE. We thank the reviewer for this precise and helpful comment. In response, we have revised the paragraph to remove references to direct exposure to green areas and instead focus on elements that are more clearly linked to indoor settings—such as views to vegetation from indoors and natural daylight exposure. This adjustment reinforces the manuscript’s focus on indoor environmental quality. The revised paragraph (Line 65) now reads:

“In contrast, other characteristics associated with indoor spaces or their immediate surroundings have been shown to foster psychological well-being. For example, having green views from indoors—such as visual access to trees or vegetation through windows—has been linked to increased perceived well-being and reduced mental fatigue (Day, 2008; Sullivan & Chang, 2011; Yuen & Nyuk Hien, 2005). Similarly, exposure to natural light within indoor spaces may alleviate symptoms of seasonal depression (Elliott et al., 1993; Matthews & Yang, 2010; Taylor et al., 1997). Overall, improvements in the indoor environment have been associated with reduced physiological and psychological distress in various contexts, including workplaces and healthcare settings (Beil & Hanes, 2013; Beukeboom et al., 2012; Codinhoto et al., 2009; Lottrup et al., 2013; Ulrich, 1981; van den Berg et al., 2010; Ward Thompson et al., 2012).”

  1. 91. I appreciate that the authors also mention positive effects of IEQ conditions. At the same time, this sentence does not clarify what these are. Air quality, ergonomics and other conditions mentioned here, are not necessarily positive. What characteristics makes them positive? Please elaborate further. In addition, why is ergonomics included here as IEQ condition, but not dealt with in the later review?

DONE. We thank the reviewer for this important clarification. To improve precision, we have revised the sentence to clarify what is meant by “positive IEQ conditions.” Additionally, we have removed “ergonomics” from the list to maintain alignment with the scope of the review and avoid introducing concepts not further addressed in the analysis. The revised paragraph (Line 77) now reads:

“The framework commonly used to conceptualize these effects is Indoor Environmental Quality (IEQ), which refers to the set of physical conditions in indoor spaces that affect occupants' health, comfort, and well-being (Abdulaali et al., 2020; Mallawaarachchi et al., 2012; Rohde et al., 2020; Steinemann et al., 2017). According to systematic reviews, IEQ dimensions such as air quality, thermal comfort, acoustic conditions, and lighting are considered positive when they help reduce discomfort or promote cognitive and emotional functioning (Al horr et al., 2016; Salonen et al., 2013). These factors have been studied in various contexts. For example, Turunen and colleagues (2014) found associations between classroom noise and poor ventilation and symptoms like fatigue and headaches in students. Similarly, Salonen and colleagues (2013) showed that adequate ventilation and favorable visual conditions—such as appropriate lighting levels—can positively affect the well-being of individuals in healthcare environments.”

  1. 117. The concept of 3E-Cognition appears to be central for the review – at least it is further mentioned in line 172. Explanations of this concept are not existing and should be added. It is also not clear, how the sentences in lines 109 to 117 relate to 3E-Cognition. I suggest to clarify these points, for example by clearing including the three key terms emergent, embodies, and extended.

DONE. The authors thank the reviewers for pointing out the need of clarification, we have added this to the text, so now the definition of each component of 3E Cognition is clearly stated. Now it reads (Line 91):

An alternative paradigm to classic conceptualizations in cognitive science conceives the nature of the cognitive process as a complex phenomenon that emerges from the dynamic relationship between the brain/body system of an agent in active interaction with its environment (Jonas, 1966; Newen et al., 2018; Thelen & Smith, 1995; Varela et al., 1991). Under this paradigm, cognition is understood as an embodied, environmentally scaffolded, and enactive process. This means: 1) that the brain and all the agent’s biology play a major role in cognition (Embodied; Thompson, 2010; Varela et al., 1991); 2) that cognitive processes are dependent on the environment and also have been transformed by environmental resources (Environmentally Scaffolded; Laland et al., 2000; Stephan & Walter, 2020; Sterelny, 2010); and 3) the mind is the product of the dynamic relationship between brain/body and environment (Enactive; Clark, 2000; Thompson, 2010).

  1. 125. This paragraph appears an unsystematic summary of some findings related to thermal aspects. How have those examples been selected and why are other IEQ-components ignored here?

DONE. We thank the reviewer for this helpful comment. We agree that the original paragraph may have appeared selective in its focus. To clarify, our intention was not to present an exhaustive summary of findings, but rather to highlight the literature on thermal environmental changes and their physiological effects as one illustrative example of methodological efforts to assess IEQ variables from an integrated, human-centered perspective. We have revised the paragraph accordingly to make this purpose clearer and to avoid the impression of an unsystematic selection.

The revised paragraph (Line 122) now reads:

Considering the aforementioned concepts, important questions arise regarding how different components of the built environment influence physiological responses. To illustrate existing methodological efforts in this area, we highlight examples from the literature on thermal environmental changes and their associations with physiological responses—such as body temperature regulation and its relationship to brain and cardiac activity. These studies exemplify how researchers are beginning to approach IEQ assessment through an inclusive lens that integrates physiological and environmental data.”

  1. 142 and in general throughout the paper. A clear definition of neuroscience research and components would be helpful. Without such clarification, readers will be confused, why measures like heart rate, EDA, skin temperature are included in this review, considering that they rather indirectly inform about cognitive processes in comparison to ECG and others. Adding a classification of the relationship between such measures and cognitive processes as well as the degree, such measures inform about cognitive processes would be crucial for readers to follow the authors line of argument.

DONE. We thank the reviewer for raising this important point. In a new footnote (Line 142), we have now explicitly defined our interpretation of "neuroscience research" asthe investigation of brain and body dynamics that directly or indirectly reflect cognitive, affective, and physiological processes”. In the revised manuscript, we clarify that while measures like heart rate, EDA, and skin temperature provide indirect indices of cognitive and emotional states through autonomic nervous system activity, they are nonetheless integral to an embodied view of cognition, as championed by 4E-Cognition frameworks (Newen et al., 2018; Parada et al., 2024; Grasso-Cladera et al., 2022). A classification table (Table 1) mapping each physiological measure to its relevance and degree of directness to cognitive processes has been added.

  1. 177. Similar to 142, it would be helpful to define the authors’ interpretation of well-being and how it related to aspects like thermal sensation and heart rate.

DONE. We appreciate the reviewers comment regarding the interpretation of well-being and the relation to physiological aspects. We believe that Table 1, as well as the following paragraph added (Line 107) in the introduction section help clarify this point:

“In the context of this review and under the 3E-Cognition framework, well-being is conceptualized as a dynamic, embodied state arising from the continuous interaction between an individual’s physiological, emotional, and cognitive processes and their environmental scaffolds. Well-being reflects the agent’s capacity to maintain adaptive regulation of bodily states, affective experiences, and cognitive engagement within specific environments. It is not a static condition, but an enacted and environmentally supported process that promotes physical comfort, emotional resilience, cognitive clarity, and effective action-in-the-world.”

  1. 194. (major) The limitation to studies that include both measures, i.e. physiological and self-reported appears a strong limitation of the approach and tremendously reducing the number of included articles. At the same time, review results presented in no way make comparisons between these two types of measures. Hence it is questionable that such inclusion criteria is justified as results would gain much more when based on a larger overview of studies in the field of IEQ and human reactions (both physiological and self-report).

DISCUSSED. We appreciate the reviewer’s thoughtful comment regarding the inclusion criteria. We agree that limiting the selection to studies that include both physiological and self-reported measures does reduce the number of eligible articles. However, we believe this approach is justified given the specific aims and scope of the present review. As outlined in the Introduction, our review seeks to systematize current scientific evidence on methodologies used to investigate the impact of the indoor built environment on well-being, with a particular focus on assessing physiological responses. Specifically, our objectives are to:

  1. Categorize the main indoor environmental quality (IEQ) variables studied in relation to well-being in real-life settings;
  2. Identify and summarize methodological aspects (e.g., data collection techniques, study settings, and biomarkers) employed in this area; and
  3. Review the self-report instruments used to assess subjective experiences of IEQ and well-being.

We recognize that several previous reviews have independently addressed the relationship between well-being and self-reported or physiological measures. However, to the best of our knowledge, no existing review has yet examined this topic from an integrated and interdisciplinary perspective—one that simultaneously considers both physiological and self-reported data. We believe this dual approach is essential for capturing the complexity of human responses to indoor environments, particularly when adopting an embodied, environmentally scaffolded, and enacted view of well-being.

To better reflect this rationale, we have now added a clarification in the manuscript to explicitly justify the inclusion criteria and their alignment with our research goals. The “Eligibility Criteria” section now reads (Line 203):

Since the present review aims to systematize the current scientific evidence on methodologies used to investigate the impact of the indoor built environment on well-being, with an emphasis on the assessment of physiological variables, only empirical studies on indoor built environment quality, incorporating physiological and well-being self-report measurements, were assessed for eligibility. 

To be included in the present review, studies should: 1) Have neurotypical human participants; 2) Study one or more of the four variables related to Indoor Environmental Quality (i.e., air quality, thermal comfort, noise, lighting); 3) Be an empirical article that addresses the relation with, at least, one physiological signal; studies only incorporating behavioral measures (e.g., only implementing eye tracking or motion energy analysis) were not considered for eligibility since the main purpose of this review is to assess physiological variables; (...)

DONE. We appreciate the comment regarding the presentation of results. We have added another column in Table 5 regarding the exploration of cross-test (i.e., an analysis that incorporates both, physiology and self report measures. We have also added a paragraph into the results section that reads (Line 459):

“Furthermore, only 6 studies explicitly conducted a cross-test between physiological responses and self-report measurements. This observation highlights a frequent methodological gap: while many studies simultaneously collected physiological and subjective data, fewer statistically analyzed their relationship.”

  1. 206. Avoid writing in verbal language style, i.e. do not use haven’t and similar phrases but write have not

DONE. We appreciate the reviewer’s observation. We have carefully revised the manuscript to replace all verbal contractions with their full forms to maintain an appropriate academic style. Specifically, we corrected the sentence:

Original: "...reports that haven't been included in a peer review process."

Revised: "...reports that have not been included in a peer review process."

  1. 216. (major) The search strategy is poorly described and the protocol on osf does not give further information. For a reproducible description, the search terms and phrases, i.e. their combination with for example “AND” or “OR” is crucial. Without such information, the method is not reproducible and the quality of the review process cannot be judged.

DONE. We thank the reviewer for this important and detailed observation. In response, we have revised the Methods section to explicitly describe the search strategy, including the use of Boolean operators and the structure of keyword combinations. We now include an example search string, and clarify that each term related to indoor environmental quality was combined individually with physiological keywords. Additionally, we have updated the protocol on OSF to include the full list of search strings for transparency and reproducibility. The revised paragraph now reads (Line 241):

“A search strategy was developed following the Peer Review of Electronic Search Strategies (PRESS) checklist (McGowan et al., 2016). The search strategy was adapted to each database. The search process was conducted by two members of the research team (AGC and MAP) after a preliminary, non-systematic review of keywords related to indoor environmental quality. The search strategy aimed to identify studies exploring the relationship between indoor environmental quality (IEQ) and physiological or neurophysiological responses. To that end, each IEQ-related term—“indoor thermal comfort”, “indoor air quality”, “indoor noise”, “indoor light”, and “indoor environment”—was combined individually with each physiological keyword: “physiolog*”, “heart*”, “brain”, “respiration”, and “skin temperature”. For example, one of the search strings was: "indoor air quality" AND "physiolog*". A full list of search strings is available in the study protocol. Keywords were searched in the articles’ abstracts. Systematics reviews and conference abstracts were excluded from the search.”

  1. 255. (major) The resulting number of included articles is very small, which – as mentioned above – might be due to a too restricted inclusion criteria or the search terms used. Such small number of articles can be justified when an in-depth analysis of their methods would be conducted. For example, going in-depth into the instruments used, their similarities and differences, which starts from scales used. In its current form, results remain superficial and narrative and do not add much value to existing literature.

DISCUSSED/ADDRESSED. The authors thank the reviewer for their feedback and the opportunity to clarify key aspects of our systematic review.

We would like to respectfully address the concern regarding the relatively small number of included studies. While the final number of articles is indeed limited, we would like to highlight that our initial search retrieved over 2,500 articles, indicating that the search strategy and terms were broad and inclusive. The reduction to a smaller set of included articles stems from the specificity of our inclusion criteria, which are closely aligned with the aims of our study and the specificity of our research question. As our review targets an emerging and highly interdisciplinary area (the intersection of environmental science and neuroscience), a more focused selection was necessary to ensure relevance and rigor.

We agree with the reviewer that a small number of included studies can be justified by conducting an in-depth analysis. In response, we have expanded our discussion to provide more detail on the methodologies employed in the selected studies, with a particular focus on the instruments and measurement scales used, highlighting both their similarities and differences. We believe this enhanced analysis strengthens the contribution of our review.

This section now reads (Line 441):

“A closer examination of the self-report instruments used across the reviewed studies reveals both convergences and divergences in methodological choices. A substantial proportion of the articles relied on scales derived from ASHRAE guidelines, reflecting a shared emphasis on thermal-related perceptions. This convergence suggests a partial standardization within the field, at least regarding thermal comfort assessments. However, significant variability was also observed. Some studies broadened their focus to include emotional states (e.g., Snow et al., 2019; Fanpu et al., 2024) or more comprehensive measures of subjective experience, such as the Positive and Negative Affect Schedule (POMS) or the Self-Assessment Manikin (SAM). Moreover, while several articles limited their evaluations to thermal parameters, others incorporated perceptions of lighting (Roy et al., 2024), air quality (Wang et al., 2023), and overall visual comfort (Fischl & Johansson, 2024). These differences highlight a lack of unified criteria for self-report measurement selection, reflecting both the multidisciplinary nature of the field and the ongoing challenges in operationalizing the construct of well-being in relation to the built environment."

Regarding the number of included articles, we are not aware of a specific threshold or standard prescribed in the existing systematic review guidelines. In fact, reviews with fewer studies are often observed in focused systematic reviews, whereas scoping reviews typically involve broader, more exploratory questions and larger sets of literature.

Finally, we would like to emphasize that, to our knowledge, this is the first review to explore the interdisciplinary integration of environmental science and neuroscience. We hope this unique perspective will be of value to the academic community and encourage further research at this intersection.

  1. 295. Table 2. Please add a footnote explaining at least the topics of Q1 to Q10, so that readers can understand the information in the table without having to look at other literature sources.

DONE. We appreciate the reviewer pointing out that the items for evaluation were missing. This was a problem with the import-export of the document. The full description of the items is available as a footnote on the table (Line 326).

Case Series Studies: Q1 Were there clear criteria for inclusion in the case series?; Q2 Was the condition measured in a standard, reliable way for all participants included in the case series?; Q3 Were valid methods used for identification of the condition for all participants included in the case series?; Q4 Did the case series have consecutive inclusion of participants?; Q5 Did the case series have complete inclusion of participants?; Q6 Was there clear reporting of the demographics of the participants in the study?; Q7 Was there clear reporting of clinical information of the participants?; Q8 Were the outcomes or follow up results of cases clearly reported?; Q9 Was there clear reporting of the presenting site(s)/clinic(s) demographic information?; Q10 Was statistical analysis appropriate?

Cross-Sectional Studies: Q1 Were the criteria for inclusion in the sample clearly defined?; Q2 Were the study subjects and the setting described in detail?; Q3 Was the exposure measured in a valid and reliable way?; Q4 Were objective, standard criteria used for measurement of the condition?; Q5 Were confounding factors identified?; Q6 Were strategies to deal with confounding factors stated?; Q7 Were the outcomes measured in a valid and reliable way?; Q8 Was appropriate statistical analysis used?

NA: not applicable

  1. 309 and other lines. Lightning is something completely different than lighting.

DONE. We thank the reviewer for this correction. We have reviewed the manuscript and replaced all incorrect uses of “lightning” with the appropriate term “lighting” when referring to indoor environmental conditions. Specifically, we corrected the sentence:

Original: "...air quality, thermal comfort, noise, lightning."

Revised: "...air quality, thermal comfort, noise, lighting."

  1. 336. (major) The definition of “real-life context”, which appears a major inclusion criteria, is not clear. It appears that many studies selected are not performed in classic laboratories, which is as targeted, while they are done in specially prepared rooms not used for daily activities. For example, this applies to Chinazzo et al which took place not in a commonly used office room, but only an office-like room specially prepared for the experiment. Similar things can be said about Zhou et al. Also Wang et al is described as “experimental room”. As such, participants of those studies are not in their normal work environment, but in a still artificial one. It would be crucial to extend the description of the settings in order to clarify the “naturalness” of these settings. Up to which point is a lab a lab? When, i.e starting from which properties, does it turn into a real-life context? What are the influences of such properties on the (ecological) validity of results? These are crucial questions that could increase the value of this review beyond the current narrative status.

DONE. The authors thank the reviewer for highlighting the need to clarify one of the main inclusion criteria regarding the definition of a "real-life context." We agree that this is a crucial conceptual point that warrants further elaboration. We have expanded our explanation of what constitutes a "real-life context" in the scope of this review, and we have clarified how this concept relates to ecological validity. Importantly, while we acknowledge that many of the included studies were not conducted in participants’ habitual environments (e.g., homes or active workplaces), the concept of ecological validity—following its established definition—is more closely tied to the nature of the task rather than the physical location of the study. Specifically, ecological validity refers to the extent to which the experimental conditions resemble real-world demands in terms of cognitive, perceptual, and behavioral processes. Thus, a study conducted in a room that simulates real-life features may still achieve high ecological validity if it reproduces functionally meaningful tasks, even if the setting itself is not a naturally occurring environment.

The section for this inclusion criteria now reads (Line 215):

Furthermore, studies will only be eligible if they are conducted in real-world scenarios (e.g., real office or residential environments, or a space accommodate to simulate the characteristics of those scenarios) instead of classical laboratory setups (i.e., participants performing a task with lower degrees of ecological validity, understood as limited representational fidelity of the stimuli and insufficient alignment of tasks with functional, goal-directed behavior (Holleman et al., 2021).

  1. 355. Why are iButtons in a different category from surface thermometers? iButtons are a brand name – truly commonly used – but at the same time, they measure surface temperature and are hence surface thermometers and not a category on their own.

DONE. We appreciate the reviewer’s clarification. We have revised the manuscript to group iButtons within the category of surface thermometers, recognizing that they are commonly used devices for measuring surface temperature rather than a distinct measurement category. This change has been reflected both in the "Data Collection Technique" section and in Table 5, where iButtons are now listed under surface thermometers to ensure consistency throughout the manuscript. The revised paragraph now reads (Line 404):

“The reviewed articles employed various data collection techniques, depending on the physiological variables of interest. For cardiac activity, the most common methods included heart rate monitors (N = 6; Barbic et al., 2022; Chinazzo et al., 2018; Fanpu et al., 2024; Fischl & Johansson, 2024; Gao et al., 2023; Wang et al., 2023), the use of sphygmomanometers (N = 3; Roy et al., 2024; Wang et al., 2018; Wang et al., 2023), pulse oximeters (N = 3; Hu et al., 2022; Roy et al., 2024; Snow et al., 2019) and blood pressure monitors (N = 2; Gao et al., 2023; Wu & Wagner, 2023). Regarding skin temperature, surface thermometers were the most frequently used devices (N = 10; Beaudette et al., 2024; Chinazzo et al., 2018, 2019; Gao et al., 2023; Hu et al., 2022; Snow et al., 2019; Song et al., 2020; Wang et al., 2018; Wu & Wagner, 2023; Zhou et al., 2023). An infrared imager was employed in one study (Gao et al., 2023). Only one article collected information on electrical brain activity using electroencephalography (EEG; Snow et al., 2019). Table 5 summarizes the instruments used and measurements conducted in the included articles.”

  1. Results in general. (major) The results section remains narrative and largely superficial. There are many more aspects related to methodological aspects which would be interesting to compare. For example, what are the experimental approaches of these studies? Are they rather observational or do they manipulate specific IEQ parameters systematically and experimentally? What is the range of conditions? What is the size of samples? And so on. Right now, the results focus on a very limited number of aspects related to complex methodological questions and has not much value.

DONE/DISCUSSED. We appreciate the reviewers comment. First, we have added to Table 3 information regarding the nature of each article (e.g., observational or with experimental manipulation of the variables) as well as sample size (Line 353 and 366). Secondly, we have discussed the comment regarding the results presentation. As described in the Methods section (Synthesis of Results), we have chosen to use a narrative approach to present our findings. This decision was primarily driven by the considerable variability across the included studies in terms of (1) experimental designs, (2) types of variables measured, and (3) reported outcomes. Due to this heterogeneity, conducting a meta-analysis—which would indeed be the most appropriate method for a quantitative synthesis—was not feasible in this case. We have now clarified this rationale more explicitly in the Methods section to ensure the reasoning behind our approach is transparent. The revised section reads as follows (Line 276):

Following a narrative methodology that Arksey and O’Malley (2005) presented, data were summarized following the data extraction categories to answer the review objectives and questions. Data will be presented graphically (graphs or diagrams) and in tables. Due to the high heterogeneity of the included articles, mainly in terms of the main goal, experimental design, and measured outcomes, a meta-analysis cannot be performed (Field et al., 2010; Soni et al., 2023).

  1. 409 and complete paragraph. (major) This part of the discussion is highly speculative. Authors appear to credit the COVID-19 pandemic leading to an increase in studies. There could be numerous other reasons for the increase, for example the general increase in publications in all fields over the last years. Hence, their argument appears not justified and needs to be made on more solid arguments and facts. For example, compare the observed increase of included studies compared to general publication trends. Also consider technological developments, which ease such studies and hence increase also the output of these. The current line of argument is too weak to be justified and ignore potential other explanations.

ADDRESSED. We appreciate the reviewer’s thoughtful observation. In response, we have revised the paragraph to avoid suggesting a direct causal link between the COVID-19 pandemic and the observed increase in studies. The updated discussion now acknowledges multiple potential contributing factors, while still reflecting on the socio-cultural contexts influencing scientific inquiry. The revised paragraph now reads (Line 479):

“From the general analysis of the research characteristics, it is possible to notice a significant increase in studies conducted in the field over the past four years (N = 11), accounting for more than 70% of the included studies. Several factors may have contributed to this trend, including the general rise in scientific publications across disciplines, technological advancements that have facilitated real-world physiological measurements, and growing attention to the role of indoor environments in health. Additionally, it is possible that the COVID-19 pandemic, by increasing the time spent indoors and reshaping the relationship between people and built environments (Daniel, 2020; Martin et al., 2022; Xiao et al., 2021) , contributed to this renewed focus. While this interpretation remains tentative, it highlights how scientific inquiry is deeply influenced by socio-cultural, political, and economic realities (Haggis, 2008; Haraway, 2020). In this sense, although the aspiration for scientific knowledge is generalizability, it remains crucial to acknowledge the specificity of contexts and the need for science to remain sensitive to diversity.” 

  1. 425. The share of studies related to location should be set in relation to overall publication trends. Chinese publications have a large general share in worldwide scientific publications in all fields.

DONE. We appreciate the reviewer’s observation. To avoid drawing unsupported interpretations and to maintain the focus of the discussion, we have removed the observation regarding the geographic concentration of studies.

  1. 445. Due to the lack of information regarding the search terms, it is not clear whether the lack of studies on noise is true or just a result of the search strategy.

ADDRESSED. We appreciate the reviewer’s observation. We have now fully detailed the search strategy in the Methods section, including the use of "indoor noise" as one of the specific keywords combined with physiological terms. Therefore, the limited number of studies related to noise identified in this review reflects the available literature matching our inclusion criteria, rather than a limitation of the search strategy. The revised paragraph in the Methods section now reads (Line 241):

“The search strategy aimed to identify studies exploring the relationship between indoor environmental quality (IEQ) and physiological or neurophysiological responses. To that end, each IEQ-related term—“indoor thermal comfort”, “indoor air quality”, “indoor noise”, “indoor light”, and “indoor environment”—was combined individually with each physiological keyword: “physiolog*”, “heart*”, “brain”, “respiration”, and “skin temperature”. For example, one of the search strings was: "indoor air quality" AND "physiolog*". A full list of search strings is available in the study protocol.”

  1. 481. When asking for the meaning of well-being, I would expect authors to reflect more on work reflecting the definition of well-being. Right now, Altomonte et al is mentioned, but there are other recent publications dealing with definitions. Why was Altomonte highlighted here and not others?

ADDRESSED. We appreciate the reviewer’s constructive comment. In response, we have enriched the paragraph by incorporating additional references addressing the conceptualization of well-being, including broader theoretical perspectives and more recent contributions relevant to the built environment. We also expanded the reflection on the complexity and variability of well-being definitions across disciplines, and their implications for measurement challenges within the built environment context. The revised paragraph now reads (Line 583):

“Well-being is widely recognized as a complex and multidimensional construct, integrating psychological, social, and physical dimensions, and influenced by the environment in which individuals find themselves (Altomonte et al., 2024; Andalib et al, 2024; Ryan et al, 2001; Scaria et al., 2020). It can be understood as a dynamic state of equilibrium that may be affected by life events and challenges (Dodge et al., 2012), encompassing dimensions such as emotional fulfillment, psychological health, and social connections (Simons & Baldwin, 2021). However, there is no international consensus on its definition, and different disciplines conceptualize well-being in varied ways. Often, the term is used simplistically as synonymous with wellness, happiness, and quality of life, or associated with comfort and health (Bluyssen et al., 2011; Ghaffarianhoseini et al., 2018; J. Lee et al., 2011; Rohde et al., 2020). 

Similarly, there is no clear consensus on what constitutes well-being in relation to the built environment. Various definitions and conceptual frameworks have been proposed, often influenced by disciplines such as architecture, environmental psychology, and public health (Altomonte et al., 2024). The complexity arises because well-being in built spaces can be affected by a broad range of factors, including environmental stimuli, individual preferences, and broader societal norms (Andalib et al., 2024; Evans, 2003). Moreover, tensions may emerge between architectural goals, such as energy efficiency, and the well-being of occupants, indicating that achieving a balance is challenging and highly context-dependent (Al horr et al., 2016). Due to the lack of consensus on the definition of well-being in the built environment, there is no straightforward approach to measuring well-being in existing or new buildings. Some studies propose that well-being should be assessed through both subjective and objective measures, acknowledging its holistic nature while differentiating it from related concepts such as health and quality of life (Andalib et al., 2024; Zhang et al., 2023). Current methods typically involve medical examinations, extensive self-report questionnaires (not exclusively focused on well-being), and diverse observational and monitoring techniques (Bluyssen et al., 2011), though these approaches rarely incorporate comparative analysis with neurological variables.”

  1. 517. The diversity of hypotheses is not part of the results section. Please consider changing to “aims”, which have been briefly touched, here or add a summary of hypotheses to the results section.

DONE. We appreciate the reviewer’s careful observation. In response, we have revised the terminology, replacing the word "hypotheses" with "aims" to more accurately reflect the nature of the information extracted from the reviewed studies. The revised sentence now reads (Line 632):
“Considering the diversity of aims, experimental designs, and technical-methodological aspects of the reviewed studies, it is difficult to determine one exclusive biomarker to address the study of the impact of the indoor built environment on well-being.”

  1. 533. It is not clear, why authors highlight HEP. Such recommendations would require much more in-depth analysis and explanation of its advantages over other methods, which at the same time should include its limitations.

ADDRESSED. We thank the reviewer for pointing out the need of more explanation on highlighting HEP. We believe that Table 1 serves to add information about the parameter that is being recommended as a possible measure to further explore the dynamic relationship between multiple systems in the body.

  1. 539 and follows. This part of the discussion appears to be disconnected from the results. Hypotheses have not been analyzed in the reviewed articles. It would be crucial to align results and discussion, hence an in-depth analysis of the current status of hypothesis-driven analysis strategies would be needed before it can be discussed here.

DONE. We appreciate the reviewer’s comment regarding the alignment between the results and the discussion. We agree that the majority of the studies included in our review did not explicitly employ hypothesis-driven methodologies, but rather exploratory or descriptive approaches. We have now clarified this point in the revised discussion. Specifically, we present our remarks on hypothesis-driven strategies not as a summary of the reviewed findings, but as a recommendation for future research in this emerging field. We have added text to make this distinction explicit, ensuring that our discussion remains grounded in the results while pointing toward methodological improvements that would strengthen future studies (Line 783): 

“Notably, across the reviewed literature, few studies employed explicit hypothesis-driven experimental designs, with most adopting exploratory or descriptive frameworks. This highlights a methodological gap that future research should address. We recommend that subsequent studies in this field increasingly implement hypothesis-driven methodologies, as they would enhance the interpretability, replicability, and theoretical development of embodied cognitive neuroscience in real-world contexts.”

  1. 562. “The importance … is increasing” This sentence and several similar statements by the authors appear to be based on their own opinion. No reference is given here to justify such statements hence it remains unknown, how authors got to such statements and the agenda behind it.
  2. 598. (major) Similar to 533, it is not clear, why MoBI is highlighted and recommended. Such recommendation is not a consequence of the results and therefore needs much more argumentation and discussion of its pros and cons compared to other approaches. Without such line of argument, it could be just the authors’ own interest.

DONE. We appreciate the reviewer’s concern and have substantially expanded the rationale for highlighting Mobile Brain/Body Imaging (MoBI). We now clarify that MoBI is proposed based on its unique capacity to simultaneously capture brain, behavior, and bodily dynamics in real-world environments, addressing a gap noted across the reviewed studies. We have discussed MoBI’s advantages (ecological validity, brain-body coupling analysis) alongside its current limitations (e.g., motion artifacts, technological complexity) throughout the manuscript. We emphasize that our recommendation stems from the identified need for more dynamic, context-sensitive methodologies in this research area.

  1. 607. Small typo By

DONE. We thank the reviewer for pointing out the typo; we have fixed it.

  1. 638. (major) Authors state here, that “it is possible to identify a significant relationship between different variables …” However, their review of the included articles does not include the results sections of those articles, solely the methodological aspects. Hence, this conclusion is not based on the results unless such analysis is added to the results section. The same statement is included in the abstract and not justified there neither.

DONE. We appreciate the reviewer's comment. We have clarified that we are pointing out an increase in the interest in the study of the association between environmental variables, well-being, and the impact on physiology. The section now reads:

Among the principal results of this review, it is possible to identify a significant interest in the relationship between different variables of the indoor built environment and psycho-physiological states, although studies of isolated variables proliferate with little holistic vision that includes multi-modal phenomena and cross-modal stimuli interaction.

  1. 642. Similar to 638, the degree of influence of IEQ on any parameter like heat rate and skin temperature is not part of the results section of this review. The results section solely looked at the number of studies that investigated such influence.

DONE. We appreciate the reviewer’s observation. In response, we have revised the phrasing to avoid implying empirical findings of influence. The corrected sentence now emphasizes that the review identified the frequency of studies investigating specific associations rather than establishing causal effects. The revised sentence now reads (Line 794):

“For instance, thermal comfort was the most commonly investigated IEQ variable in relation to heart activity and skin temperature.”

  1. 645. As mentioned above, the justification of the recommendation for MoBI is not convincing in the discussion section and needs to be improved.

DONE. We appreciate the reviewer’s suggestion, we have added new text justifying MoBI (Line 730): 

“MoBI represents a methodological advancement, uniquely suited to addressing key gaps identified across the reviewed literature (Gramann 2024). Specifically, MoBI enables the simultaneous recording of neural, behavioral, and bodily dynamics in real-world or ecologically valid environments. This approach directly aligns with the embodied, environmentally scaffolded, and enacted principles of the 3E-Cognition framework that grounds this review. While traditional laboratory-based neuroscience often isolates cognitive processes from natural contexts, MoBI captures the brain-body-environment system as it dynamically unfolds (Stangl et al., 2023). The advantages of MoBI include the ability to study active cognition and behavior outside of constrained laboratory settings (Parada 2018), providing critical insights into brain-body coupling mechanisms during naturalistic interaction (Costa-Cordella et al., 2024; Grasso-Cladera et al., 2023; Jacobsen et al., 2025). Moreover, it allows researchers to explore how environmental features shape neurobehavioral dynamics, a dimension that static or purely lab-based methods cannot adequately capture.

Nonetheless, it is important to acknowledge MoBI’s current limitations, including the potential presence of motion artifacts (Klug and Gramann, 2024), increased data complexity, and the technological challenges associated with portable neuroimaging. These issues, however, are being progressively addressed through advances in signal processing, hardware design, and hybrid methodological frameworks.

Thus, our recommendation for increased use of MoBI stems not from personal preference, but from the necessity to implement more dynamic, context-sensitive methodologies that are theoretically and empirically suited to studying cognition as it naturally occurs in lived environments.”

Reviewer 3 Report

Comments and Suggestions for Authors

1. This study notes that "noise was not explicitly addressed in any of the included studies." However, substantial empirical evidence supports noise's impact on psychological stress, autonomic nervous activity, sleep quality, and learning performance. Could the search strategy or exclusion criteria have overly emphasized physiological signals, thereby overlooking studies focused on noise that otherwise met inclusion standards?

2. Human responses to the environment are often the result of multisensory integration and interactive effects. Should future research develop composite IEQ models incorporating interaction analyses, such as "thermal × natural light" or "thermal comfort × air quality," to reflect real-life complexity better?

3.Many global healthy building frameworks—such as the IWBI WELL standard—align indoor environmental factors with human health and comfort. For example, lighting conditions are evaluated based on Equivalent Melanopic Lux (EML). This aspect warrants further integration of similar healthy building research to enrich the discussion.

Author Response

Reviewer 3:

  1. This study notes that "noise was not explicitly addressed in any of the included studies." However, substantial empirical evidence supports noise's impact on psychological stress, autonomic nervous activity, sleep quality, and learning performance. Could the search strategy or exclusion criteria have overly emphasized physiological signals, thereby overlooking studies focused on noise that otherwise met inclusion standards?

DONE. We appreciate the reviewer's comment regarding the selected keywords' limitations for doing the searches and their impact on the included articles. We have added a point to the limitations section addressing this issue (Line 763):

“First, no eligible articles in the reviewed literature included noise as an Indoor Environmental Quality (IEQ) variable, despite its recognized importance in influencing well-being and comfort in indoor environments. This gap highlights the need for more comprehensive research incorporating noise alongside other IEQ variables to better understand its impact on physiological and psychological states. However, it is important to consider that the conceptualization of noise for the searches might have been a limitation on the number and type of articles.

  1. Human responses to the environment are often the result of multisensory integration and interactive effects. Should future research develop composite IEQ models incorporating interaction analyses, such as "thermal × natural light" or "thermal comfort × air quality," to reflect real-life complexity better?

DONE. The authors thank the reviewer for the insightful comments regarding the development of more multidimensional IEQ models. We have added an entire paragraph in our discussion section addressing this matter, which reads (Line 536):

“Nevertheless, human responses to indoor environmental quality (IEQ) are inherently multisensory and shaped by the dynamic integration of multiple environmental cues. Rather than being driven by isolated parameters, psychological and physiological responses often emerge from the interplay between sensory modalities such as thermal, visual, olfactory, and auditory stimuli. For instance, studies have shown that the perception of thermal comfort can be modulated by lighting conditions, with natural light enhancing tolerance to warmer temperatures (Li et al., 2023; Azzazy et al., 2024). Similarly, air quality may interact with thermal environments to influence cognitive performance and perceived well-being (Wargocki & Wyon, 2013; Zhang et al., 2017). This highlights the need for future research to move beyond unidimensional models and develop composite IEQ frameworks that account for interaction effects—e.g., thermal × natural light or thermal comfort × air quality—to better capture the complexity of real-life environmental experiences. Such integrative approaches are particularly crucial in neuroscientific investigations where multisensory convergence plays a key role in shaping emotional, cognitive, and behavioral outcomes (Spence, 2023). Embracing this complexity will enhance built environment research's ecological validity and translational value.”

  1. Many global healthy building frameworks—such as the IWBI WELL standard—align indoor environmental factors with human health and comfort. For example, lighting conditions are evaluated based on Equivalent Melanopic Lux (EML). This aspect warrants further integration of similar healthy building research to enrich the discussion.

DONE. We appreciate the reviewer’s insightful and constructive suggestion. In response, we have expanded the discussion to acknowledge the contribution of healthy building frameworks, such as the WELL Building Standard, in operationalizing the relationship between indoor environmental quality factors and human health, comfort, and well-being outcomes. The added paragraph reads (Line 560):

“Moreover, it is worth noting that several global healthy building frameworks, such as the WELL Building Standard (International WELL Building Institute, 2022), have explicitly aligned indoor environmental factors with human health, comfort, and well-being outcomes. For example, lighting conditions are evaluated using metrics like Equivalent Melanopic Lux (EML) to account for circadian and physiological effects. Recent studies have shown that WELL-certified and LEED-certified buildings tend to exhibit improved indoor environmental quality parameters compared to non-certified buildings, highlighting the tangible link between building standards and occupant well-being (Kent et al., 2024). Additionally, certifications such as WELL incorporate multi-criteria evaluation methods that reflect a holistic view of human well-being in relation to the built environment (Nicolini, 2022). The growing alignment between healthy building certifications and scientific research on IEQ underscores the importance of adopting interdisciplinary and ecologically valid approaches when assessing the impacts of the built environment on occupants.”

Round 2

Reviewer 2 Report

Comments and Suggestions for Authors

The authors have greatly improved their manuscript and addressed many comments of this reviewer sufficiently. A few points remain, that would need further clarification or revision.

As row numbers are missing in the available pdf version, previous line comments are used as a basis to continue the discussion where needed marked by “V2” at the beginning of the comment.

  1. First, the search strategy is not reproducible as the chosen search terms and phrases are not reported. Hence, I cannot judge the suitability of these and their implications for the review process. A final assessment of the quality of the methodology is therefore not possible.

DONE.  We appreciate the reviewer’s critical observation. To ensure reproducibility and transparency, we have revised the Methods section to report the full list of keywords used, and clarify how the search terms were combined using Boolean logic. We have also included an example search string and noted that the complete list of queries is available in the study protocol (updated). The revised paragraph now reads (Line 241):

“A search strategy was developed following the Peer Review of Electronic Search Strategies (PRESS) checklist (McGowan et al., 2016). The search strategy was adapted to each database. The search process was conducted by two members of the research team (AGC and MAP) after a preliminary, non-systematic review of keywords related to indoor environmental quality. The search strategy aimed to identify studies exploring the relationship between indoor environmental quality (IEQ) and physiological or neurophysiological responses. To that end, each IEQ-related term—“indoor thermal comfort”, “indoor air quality”, “indoor noise”, “indoor light”, and “indoor environment”—was combined individually with each physiological keyword: “physiolog*”, “heart*”, “brain”, “respiration”, and “skin temperature”. For example, one of the search strings was: "indoor air quality" AND "physiolog*". A full list of search strings is available in the study protocol. Keywords were searched in the articles’ abstracts. Systematics reviews and conference abstracts were excluded from the search.”

V2. Thank you for adding the search strings and more details regarding the search strategy. The method is now reproducible. At the same time, the chosen search strings demonstrate a strong limitation of this study, that should be generally stated at least in the limitations section – not only for noise. The issue related to the combination of “indoor” together with a domain. Such combination might ensure that only studies related to indoors are addressed, but at the same time limits the search results strongly. Many researchers in the field of IEQ do not necessarily combine their domain description with “indoors”. In addition, domains appear to be treated differently. For the thermal domain, the search string requires “comfort” being part of the string, while for other domains, such as air quality, noise and light, such requirement does not exist. A single wording for IEQ terms without wild cards, i.e. * as it was done for physiological terms further reduces the number of articles found. For example, lighting would not be found. Studies referring to temperature instead of thermal would not be found. Very important, the term “noise” itself is less frequently used compared to “sound” or “acoustic” in the corresponding domain. As such – also based on the reviewers randomly checked search results reflecting above aspects, which led to much larger initial figures of detected articles, there is a likelihood that the low number of identified studies is due to the search strategy and not reflecting the field. Authors need to either repeat their review with a more proper search term selection or strongly acknowledge this major limitation prominently throughout their article.

  1. 255. (major) The resulting number of included articles is very small, which – as mentioned above – might be due to a too restricted inclusion criteria or the search terms used. Such small number of articles can be justified when an in-depth analysis of their methods would be conducted. For example, going in-depth into the instruments used, their similarities and differences, which starts from scales used. In its current form, results remain superficial and narrative and do not add much value to existing literature.

DISCUSSED/ADDRESSED. The authors thank the reviewer for their feedback and the opportunity to clarify key aspects of our systematic review.

We would like to respectfully address the concern regarding the relatively small number of included studies. While the final number of articles is indeed limited, we would like to highlight that our initial search retrieved over 2,500 articles, indicating that the search strategy and terms were broad and inclusive. The reduction to a smaller set of included articles stems from the specificity of our inclusion criteria, which are closely aligned with the aims of our study and the specificity of our research question. As our review targets an emerging and highly interdisciplinary area (the intersection of environmental science and neuroscience), a more focused selection was necessary to ensure relevance and rigor.

V2. The initial number of 2,500 articles is not reflecting a broad search, with many reviews these days starting with 10 to 20,000 articles.

We agree with the reviewer that a small number of included studies can be justified by conducting an in-depth analysis. In response, we have expanded our discussion to provide more detail on the methodologies employed in the selected studies, with a particular focus on the instruments and measurement scales used, highlighting both their similarities and differences. We believe this enhanced analysis strengthens the contribution of our review.

This section now reads (Line 441):

“A closer examination of the self-report instruments used across the reviewed studies reveals both convergences and divergences in methodological choices. A substantial proportion of the articles relied on scales derived from ASHRAE guidelines, reflecting a shared emphasis on thermal-related perceptions. This convergence suggests a partial standardization within the field, at least regarding thermal comfort assessments. However, significant variability was also observed. Some studies broadened their focus to include emotional states (e.g., Snow et al., 2019; Fanpu et al., 2024) or more comprehensive measures of subjective experience, such as the Positive and Negative Affect Schedule (POMS) or the Self-Assessment Manikin (SAM). Moreover, while several articles limited their evaluations to thermal parameters, others incorporated perceptions of lighting (Roy et al., 2024), air quality (Wang et al., 2023), and overall visual comfort (Fischl & Johansson, 2024). These differences highlight a lack of unified criteria for self-report measurement selection, reflecting both the multidisciplinary nature of the field and the ongoing challenges in operationalizing the construct of well-being in relation to the built environment."

V2. Please check abbreviation POMS. This reviewer is more familiar with PANAS for the Positive and Negative Affect Schedule. POMS is Profile of Mood State

  1. 336. (major) The definition of “real-life context”, which appears a major inclusion criteria, is not clear. It appears that many studies selected are not performed in classic laboratories, which is as targeted, while they are done in specially prepared rooms not used for daily activities. For example, this applies to Chinazzo et al which took place not in a commonly used office room, but only an office-like room specially prepared for the experiment. Similar things can be said about Zhou et al. Also Wang et al is described as “experimental room”. As such, participants of those studies are not in their normal work environment, but in a still artificial one. It would be crucial to extend the description of the settings in order to clarify the “naturalness” of these settings. Up to which point is a lab a lab? When, i.e starting from which properties, does it turn into a real-life context? What are the influences of such properties on the (ecological) validity of results? These are crucial questions that could increase the value of this review beyond the current narrative status.

DONE. The authors thank the reviewer for highlighting the need to clarify one of the main inclusion criteria regarding the definition of a "real-life context." We agree that this is a crucial conceptual point that warrants further elaboration. We have expanded our explanation of what constitutes a "real-life context" in the scope of this review, and we have clarified how this concept relates to ecological validity. Importantly, while we acknowledge that many of the included studies were not conducted in participants’ habitual environments (e.g., homes or active workplaces), the concept of ecological validity—following its established definition—is more closely tied to the nature of the task rather than the physical location of the study. Specifically, ecological validity refers to the extent to which the experimental conditions resemble real-world demands in terms of cognitive, perceptual, and behavioral processes. Thus, a study conducted in a room that simulates real-life features may still achieve high ecological validity if it reproduces functionally meaningful tasks, even if the setting itself is not a naturally occurring environment.

The section for this inclusion criteria now reads (Line 215):

Furthermore, studies will only be eligible if they are conducted in real-world scenarios (e.g., real office or residential environments, or a space accommodate to simulate the characteristics of those scenarios) instead of classical laboratory setups (i.e., participants performing a task with lower degrees of ecological validity, understood as limited representational fidelity of the stimuli and insufficient alignment of tasks with functional, goal-directed behavior (Holleman et al., 2021).

V2. Authors definition of ecological validity is correct, “ecological” was only in brackets in previous comment. Looking for example at the broader concept of external validity, there are likely larger differences that will reduce generalizability of the results obtained with the studies described. The latest additions by the authors explain their rational with respect to ecological validity. At the same time, adding a comment on the external validity to limitations could be meaningful.

  1. 409 and complete paragraph. (major) This part of the discussion is highly speculative. Authors appear to credit the COVID-19 pandemic leading to an increase in studies. There could be numerous other reasons for the increase, for example the general increase in publications in all fields over the last years. Hence, their argument appears not justified and needs to be made on more solid arguments and facts. For example, compare the observed increase of included studies compared to general publication trends. Also consider technological developments, which ease such studies and hence increase also the output of these. The current line of argument is too weak to be justified and ignore potential other explanations.

ADDRESSED. We appreciate the reviewer’s thoughtful observation. In response, we have revised the paragraph to avoid suggesting a direct causal link between the COVID-19 pandemic and the observed increase in studies. The updated discussion now acknowledges multiple potential contributing factors, while still reflecting on the socio-cultural contexts influencing scientific inquiry. The revised paragraph now reads (Line 479):

“From the general analysis of the research characteristics, it is possible to notice a significant increase in studies conducted in the field over the past four years (N = 11), accounting for more than 70% of the included studies. Several factors may have contributed to this trend, including the general rise in scientific publications across disciplines, technological advancements that have facilitated real-world physiological measurements, and growing attention to the role of indoor environments in health. Additionally, it is possible that the COVID-19 pandemic, by increasing the time spent indoors and reshaping the relationship between people and built environments (Daniel, 2020; Martin et al., 2022; Xiao et al., 2021) , contributed to this renewed focus. While this interpretation remains tentative, it highlights how scientific inquiry is deeply influenced by socio-cultural, political, and economic realities (Haggis, 2008; Haraway, 2020). In this sense, although the aspiration for scientific knowledge is generalizability, it remains crucial to acknowledge the specificity of contexts and the need for science to remain sensitive to diversity.” 

V2. Was COVID-19 really increasing the time spent indoors, or only the time spent at home (compared to other indoor environments)?

  1. 445. Due to the lack of information regarding the search terms, it is not clear whether the lack of studies on noise is true or just a result of the search strategy.

ADDRESSED. We appreciate the reviewer’s observation. We have now fully detailed the search strategy in the Methods section, including the use of "indoor noise" as one of the specific keywords combined with physiological terms. Therefore, the limited number of studies related to noise identified in this review reflects the available literature matching our inclusion criteria, rather than a limitation of the search strategy. The revised paragraph in the Methods section now reads (Line 241):

“The search strategy aimed to identify studies exploring the relationship between indoor environmental quality (IEQ) and physiological or neurophysiological responses. To that end, each IEQ-related term—“indoor thermal comfort”, “indoor air quality”, “indoor noise”, “indoor light”, and “indoor environment”—was combined individually with each physiological keyword: “physiolog*”, “heart*”, “brain”, “respiration”, and “skin temperature”. For example, one of the search strings was: "indoor air quality" AND "physiolog*". A full list of search strings is available in the study protocol.”

V2. See first comment, especially related to the use of “noise” as single search string for the acoustic domain.

  1. 539 and follows. This part of the discussion appears to be disconnected from the results. Hypotheses have not been analyzed in the reviewed articles. It would be crucial to align results and discussion, hence an in-depth analysis of the current status of hypothesis-driven analysis strategies would be needed before it can be discussed here.

DONE. We appreciate the reviewer’s comment regarding the alignment between the results and the discussion. We agree that the majority of the studies included in our review did not explicitly employ hypothesis-driven methodologies, but rather exploratory or descriptive approaches. We have now clarified this point in the revised discussion. Specifically, we present our remarks on hypothesis-driven strategies not as a summary of the reviewed findings, but as a recommendation for future research in this emerging field. We have added text to make this distinction explicit, ensuring that our discussion remains grounded in the results while pointing toward methodological improvements that would strengthen future studies (Line 783): 

“Notably, across the reviewed literature, few studies employed explicit hypothesis-driven experimental designs, with most adopting exploratory or descriptive frameworks. This highlights a methodological gap that future research should address. We recommend that subsequent studies in this field increasingly implement hypothesis-driven methodologies, as they would enhance the interpretability, replicability, and theoretical development of embodied cognitive neuroscience in real-world contexts.”

V2. Please avoid terms such as “few” as in “few studies” and replace these with exact numbers.

Author Response

Comment 1: 

We thank the reviewer for this insightful observation. We fully acknowledge that our search strategy, while methodologically transparent and reproducible, may have inadvertently restricted the scope of captured literature due to the use of specific terms like “indoor” in combination with each IEQ domain, inconsistent qualifiers across domains, and the absence of lexical variations or wildcards. This likely contributed to the relatively low number of identified studies, particularly for certain IEQ domains. Considering this, we have included this as a “major methodological limitation” in the revised manuscript to honor your insightful observation. Also we have clarified that our findings reflect studies captured under a constrained lexical scope. This acknowledgement highlights the need for future reviews to incorporate broader, more inclusive term sets -potentially including wildcards and alternative formulations (e.g., “acoustic”, “Noise”, “lightning”, “light”, and so on). 

We think this transparency strengthens our work by highlighting the challenges of balancing specificity and breadth in systematic reviews, particularly within emerging and complex interdisciplinary topics and fields such as neuroarchitecture and environmental neuroscience. The updated text reads as follows: 

While our review followed established systematic guidelines and aimed for methodological transparency, we acknowledge that the specificity of our search strategy may have unintentionally limited the breadth of included studies. In particular, the requirement to combine the term “indoor” with each Indoor Environmental Quality (IEQ) domain (e.g., “indoor noise,” “indoor lighting”) may have excluded relevant studies that examine environmental conditions within buildings but do not explicitly use the term “indoor.” Furthermore, the inclusion criteria applied inconsistently across domains—for example, “thermal comfort” was used as a compound term, while other domains like “air quality,” “noise,” and “light” were included in simpler forms. Additionally, we did not use wildcards or lexical variants (e.g., “light*” to capture “light” and “lighting,” or “sound” and “acoustic” as alternatives to “noise”), which may have further reduced the number of retrieved articles. As a result, the relatively low number of included studies in certain domains may not accurately reflect the overall research activity in the field, but rather a consequence of our constrained lexical strategy. We see this as a critical limitation of the current review and encourage future systematic reviews in this domain to employ broader, more inclusive term sets and consider piloting alternative search strategies to improve coverage and sensitivity.

Comment 2: 

We agree with the reviewer that a small number of included studies can be justified by conducting an in-depth analysis. In response, we have expanded our discussion to provide more detail on the methodologies employed in the selected studies, with a particular focus on the instruments and measurement scales used, highlighting both their similarities and differences. We believe this enhanced analysis strengthens the contribution of our review.

This section now reads (Line 441):

“A closer examination of the self-report instruments used across the reviewed studies reveals both convergences and divergences in methodological choices. A substantial proportion of the articles relied on scales derived from ASHRAE guidelines, reflecting a shared emphasis on thermal-related perceptions. This convergence suggests a partial standardization within the field, at least regarding thermal comfort assessments. However, significant variability was also observed. Some studies broadened their focus to include emotional states (e.g., Snow et al., 2019; Fanpu et al., 2024) or more comprehensive measures of subjective experience, such as the Profile of Mood States (POMS) or the Self-Assessment Manikin (SAM). Moreover, while several articles limited their evaluations to thermal parameters, others incorporated perceptions of lighting (Roy et al., 2024), air quality (Wang et al., 2023), and overall visual comfort (Fischl & Johansson, 2024). These differences highlight a lack of unified criteria for self-report measurement selection, reflecting both the multidisciplinary nature of the field and the ongoing challenges in operationalizing the construct of well-being in relation to the built environment."

Comment 3:  The authors thank the reviewer for catching this error. Effectively we meant POMS, so we have adjusted the text.

Comment 4: 

We thank the reviewer for the clarification and the valuable suggestion. We agree that while our focus on ecological validity aligns with the aims of this review, issues concerning external validity—particularly regarding the generalizability of findings across broader populations and contexts—remain important. We have therefore included an explicit comment on external validity in the Limitations section to highlight this consideration and strengthen the transparency of our conclusions. The added text reads as follows:

In addition, while our focus on ecological validity aimed to capture studies conducted in real-world contexts, we recognize that the broader concept of external validity—particularly the generalizability of findings across populations, settings, and cultures—may be limited. The heterogeneity of study designs, participant characteristics, and contextual variables in the included studies poses challenges for drawing generalized conclusions. This limitation further supports the need for standardized methodologies and cross-context replications in future research.”

Comment 5: 

We thank the reviewer for this thoughtful observation. We agree that during the COVID-19 pandemic, the increase in time spent indoors primarily reflected increased time at home, rather than across all indoor environments. We have revised the manuscript to clarify this distinction and ensure our wording more accurately reflects the nature of this shift in indoor occupancy patterns. 

Comment 6: See response to comment 1.

Comment 7: 

We appreciate the reviewer’s emphasis on this point. We agree that limiting our search to the term “noise” may have excluded studies that use alternative terms such as “sound” or “acoustic.” This lexical restriction likely contributed to the underrepresentation of the acoustic domain in our results. We have now addressed this in the revised Limitations section, where we acknowledge the need for more inclusive and semantically broad search strategies in future reviews.

Reviewer 3 Report

Comments and Suggestions for Authors

The article has many references. The format and content of the references can be revised by referring to the journal.

Author Response

We thank the reviewer for the observation. Given the interdisciplinary scope and the breadth of topics covered in this review, we consider the current number of references necessary to support the manuscript’s conceptual and methodological foundations. However, we are of course happy to adjust the number or formatting of references at the discretion of the editor.